# Epstein–Barr virus particles induce centrosome amplification and chromosomal instability

Anatoliy Shumilov[1,2,3,*], Ming-Han Tsai[1,2,3,*], Yvonne T. Schlosser[4], Anne-Sophie Kratz[4], Katharina Bernhardt[1,2,3], Susanne Fink[1,2,3], Tuba Mizani[1,2,3], Xiaochen Lin[1,2,3], Anna Jauch[5], Josef Mautner[6,7,8], Annette Kopp-Schneider[9], Regina Feederle[1,2,10], Ingrid Hoffmann[4] & Henri-Jacques Delecluse[1,2,3]

Infections with Epstein–Barr virus (EBV) are associated with cancer development, and EBV lytic replication (the process that generates virus progeny) is a strong risk factor for some cancer types. Here we report that EBV infection of B-lymphocytes (*in vitro* and in a mouse model) leads to an increased rate of centrosome amplification, associated with chromosomal instability. This effect can be reproduced with virus-like particles devoid of EBV DNA, but not with defective virus-like particles that cannot infect host cells. Viral protein BNRF1 induces centrosome amplification, and BNRF1-deficient viruses largely lose this property. These findings identify a new mechanism by which EBV particles can induce chromosomal instability without establishing a chronic infection, thereby conferring a risk for development of tumours that do not necessarily carry the viral genome.

[1] German Cancer Research Centre (DKFZ), Unit F100, 69120 Heidelberg, Germany. [2] Inserm unit U1074, DKFZ, 69120 Heidelberg, Germany. [3] German Centre for Infection Research (DZIF), 69120 Heidelberg, Germany. [4] German Cancer Research Centre (DKFZ), Unit F045, 69120 Heidelberg, Germany. [5] Institute of Human Genetics University Hospital Heidelberg, 69120 Heidelberg, Germany. [6] Helmholtz Zentrum München, Research Unit Gene Vectors, 81377 Munich, Germany. [7] Children's Hospital Technische Universität München, 80804 Munich, Germany. [8] German Center for Infection Research (DZIF), 81377 Munich, Germany. [9] German Cancer Research Centre (DKFZ), Unit C060, 69120 Heidelberg, Germany. [10] Helmholtz Zentrum München, German Research Center for Environmental Health, Institute for Diabetes and Obesity, Core Facility Monoclonal Antibodies, 81377 Munich, Germany. * These authors contributed equally to this work. Correspondence and requests for materials should be addressed to H.-J.D. (email: h.delecluse@dkfz.de).

The large majority of the world population is infected by the Epstein–Barr virus (EBV) that establishes a lifelong infection, usually without clinical consequences[1]. However, EBV infection is etiologically associated with the development of up to 2% of all human cancers[2,3]. EBV is endowed with powerful transforming abilities that are promptly revealed upon infection of B cells, its main target[1]. Three days after infection, B cells initiate cell division and readily establish permanently growing cell lines, termed lymphoblastoid cell lines (LCLs)[1]. This phenomenon can also be observed *in vivo*, for example, in infectious mononucleosis syndromes during which EBV-infected B-cell blasts proliferate in the peripheral blood and the lymph nodes of infected individuals[4,5]. These proliferating B cells can also give rise to a tumour in immunocompromised patients, in particular in transplant recipients who receive immunosuppressive drugs[6]. EBV-mediated transformation requires the simultaneous expression of most latent proteins that belong to the LMP1 and EBNA families. BHRF1, an antiapoptotic viral homologue of the Bcl2 protein and EBV microRNAs also markedly modulate this process[1,7–10]. However, most tumours induced by the virus do not express all latent genes and all EBV miRNAs[1,11,12]. Although proteins such as EBNA1, LMP1 and LMP2 or the BART miRNAs have been shown to contribute to the acquisition of the malignant phenotype in EBV-associated nasopharyngeal and gastric carcinomas or in Hodgkin's disease and Burkitt's lymphomas, the precise contribution of the virus to the transformation process in these cases remains unclear[1].

Epidemiological studies have shown that lytic replication, the process that generates new virus progeny in infected cells, is a risk factor for cancer development. High antibody titres against viral proteins that are expressed during virus lytic replication are predictive of nasopharyngeal carcinoma[13,14]. Other environmental risk factors for this tumour, such as the consumption of nitrosamines or phorbol esters in food or smoking, have all been shown to activate EBV lytic replication[15–18].

In this paper, we show that EBV lytic replication has a marked influence on the genetic stability of infected cells.

## Results

**EBV replication in B cells increases chromosomal instability.** We addressed the contribution of EBV lytic replication to the neoplastic process induced by the virus by comparing B cells infected with the highly replicating strain M81 that was isolated from a nasopharyngeal carcinoma and a replication-deficient mutant thereof (M81/ΔZR). We began our investigations by comparing the mitoses of cells either stimulated with pokeweed mitogen (PWM) or infected with either M81 or M81/ΔZR. At day 3 post treatment, dividing PWM-stimulated B cells displayed typical mitotic figures at different stages, with equal distribution of chromosomes in daughter cells (Supplementary Fig. 1a–c). In contrast, many dividing cells infected with either type of virus exhibited abnormal mitoses. Some mitoses were multipolar, others were bipolar but arranged around multiple centrioles (Figs 1a,b and 2a). Some mitoses contained non-aligned chromosomes and some anaphases showed images of chromosome lagging (Fig. 1c,d). We also found asymmetrical anaphases in which the chromosome sets were imperfectly distributed (Fig. 1e). Altogether, this set of experiments showed that 15 to 42% of mitoses in infected cells displayed an abnormal organization, which compares to 0 to 6% after PWM stimulation (Fig. 1j). Moreover, 2.2 to 7% of interphase cells showed more than four centrioles in the virus-infected population (Figs 1f and 2b).

Six days after infection, the cells with abnormal nuclei became also visible. Some cells displayed two to four equally sized nuclei,

others carried one or several micronuclei coexisting with a nucleus of approximately normal size (Figs 1g,h and 2c,d). Other cells contained a single large nucleus that proved to be polyploid after staining with serum from CREST patients that evidences the number of centromeres (Fig. 1i). Giemsa staining of mitotic plates showed that 25 to 40% of cells in these samples were aneuploid and up to 3% were polyploid (Fig. 2e,f). We performed multiplex fluorescence *in situ* hybridization (M-FISH) on three sample pairs 6 days after infection with M81 or M81/ΔZR (Supplementary Fig. 2). This analysis confirmed the high level of aneuploidy in cells infected with either type of viruses (average 29.2%), but also the presence of rare cells with chromosome deletions (2/120) or translocations (3/120). However, none of these abnormalities were clonal, that is, found in more than two mitoses of the same sample. At this time point, PWM-stimulated cells had died and could not be analysed. We continued to monitor the cells infected with M81 and M81/ΔZR until day 30 postinfection, when lytic replication begins in cells infected with wild-type viruses. By then, both centrosomal amplification and aneuploidy rates had been reduced by approximately 3-fold in cells infected with M81/ΔZR, implying that the conditions that led to their appearance vanished over time (Fig. 2a,b,e). The investigation of cells infected with M81/ΔZR at day 3, 6, 15 and 30 post infection showed a regular decrease in the rate of centrosome amplification (Supplementary Fig. 3). In contrast, although cells infected with the wild-type virus showed an initial decrease in the percentage of cells showing centrosome amplification, this rate sharply re-increased at day 30 when infected cells start to replicate (Fig. 2a,b, Supplementary Fig. 3a,b).

M-FISH karyotyping of four sample pairs confirmed the much higher level of aneuploidy in cells infected with the wild-type virus than in those infected with the replication-deficient mutant after 30 days of infection (average 38.75 versus 9%) (Fig. 3, Supplementary Fig. 4). The former cells also more frequently carried structural rearrangements, including chromosome deletions and translocations. Two of these four samples infected with wild type but none of those infected with M81/ΔZR showed a clonal abnormality, defined by more than two identical abnormal mitoses for structural abnormalities and more than three mitoses for chromosome loss. One B-cell sample infected with wild-type virus carried a recurrent t(6;9), the other showed a clonal loss of the chromosome Y (Supplementary Fig. 4). We extended our observations to cells infected with B95-8, a virus strain that hardly induces lytic replication, and found that they exhibited a pattern of chromosomal instability (CIN) and aneuploidy very similar to the one induced by M81/ΔZR (Supplementary Figs. 1d–i, 3c,d and 4b,d,h). We also analysed a cell line infected by B95-8 using M-FISH 60 days after infection and found that it carried a recurrent t(9;15) (Supplementary Fig. 4d,h).

**EBV infection induces chromosomal instability *in vivo*.** We then injected resting primary B cells briefly exposed *in vitro* to EBV into immunodeficient NSG mice. Although infection of resting B cells with the wild-type or with replication-deficient viruses gave rise to an identical rate of cell transformation and cell growth rate *in vitro*, intraperitoneal injection of $4 \times 10^4$ B cells infected with M81 wild type gave rise to tumour development more frequently than infection with the replication-deficient mutant (Fig. 4a–c). This difference in incidence disappeared after the injection of ten times more ($4 \times 10^5$) EBV-infected cells. However, in that case, the tumour burden developed by the animals was higher after infection with wild-type virus (Fig. 4d). Immunohistochemical analysis of the tumour samples confirmed that the tumour cells were infected by EBV, and that only cells infected with the wild-type virus underwent lytic replication

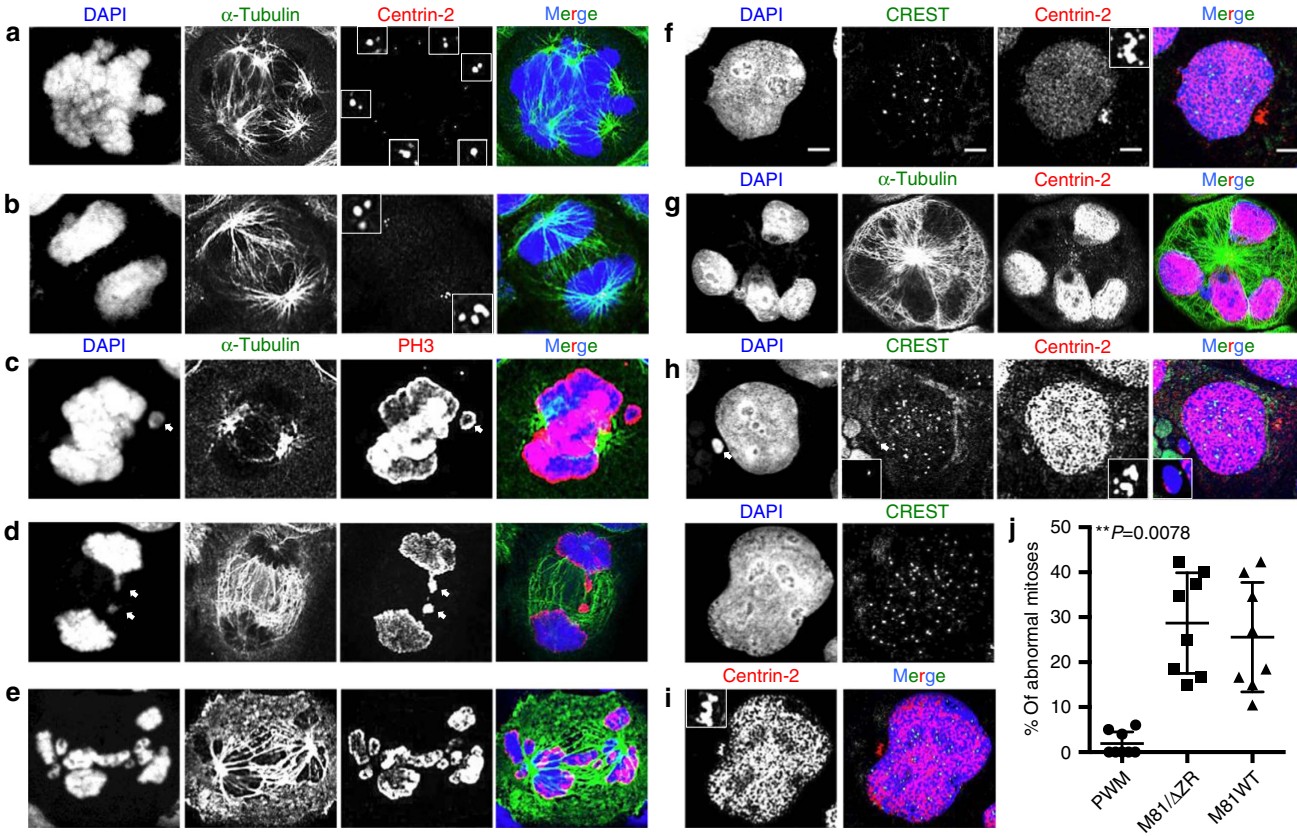

**Figure 1 | B cells infected by the Epstein–Barr virus display features of chromosomal instability.** The cells were kept in culture for 3 or 6 days after infection, cytospinned and stained for α-tubulin, centrin-2, PH3, a marker of mitotic chromosomes, or CREST, a marker of centromeres. We report the analysis of eight blood samples. For each sample, at least 100 mitoses and 500 interphase cells from cytospinned infected cells were examined. Scale bar, 5 μm (**a**) Cell undergoing a multipolar mitosis organized around six centrosomes. (**b**) Cell in anaphase organized around an increased number of centrioles. (**c**) The picture shows a non-aligned chromosome (arrow) in a cell undergoing metaphase. (**d**) This cell in anaphase shows two lagging chromosomes (arrows). (**e**) Mitotic cell showing asymmetric partition of the chromosomes. (**f**) Interphase cells with an increased number of centrioles. The inset shows a magnified view of centrosomes. (**g**) Cell with multiple nuclei. (**h**) Interphase cell that displays a micronucleus next to a larger nucleus, as well as multiple centrosomes that are magnified in the inset. (**i**) Polyploid cell with a single nucleus containing more than 46 centromeres. (**j**) The dot plot shows a summary of the frequency of abnormal mitoses identified with the stains described in **a**–**h** in B cells from the same individual stimulated with pokeweed mitogen or infected with wild-type M81 or M81/ΔZR. This analysis excludes the frequency of aneuploidy described in the sequel. Some of the obtained results included null values. Therefore, we applied an exact Wilcoxon signed-rank test to compare the results ($P = 0.0078$). Error bars represent the mean with s.d.

(Fig. 4e). The frequency of aneuploidy and centrosomal abnormalities in these tumours was two to three times higher after infection with wild-type viruses relative to the M81/ΔZR mutant, and the absolute frequency of many of these abnormalities was higher than those observed *in vitro* (compare Figs 2 and 5).

**EBV infection induces centrosome overduplication.** Centrosome amplification can result from a centrosome overduplication during the S phase or from centrosome accumulation that takes place after mitotic slippage, when dividing cells revert to the G1 phase without partitioning their chromosomes, thereby becoming tetraploid and equipped with two centrosomes[19]. We investigated both possibilities by staining cells with an increased number of centrosomes with an antibody against the CEP170 protein that associates with subdistal appendages of mother centrioles[20] (Fig. 6a–d). Centriole overduplication gives rise to a higher number of daughter centrioles than of mother centrioles, whereas centriole accumulation gives rise to an equal number of mother and daughter centrioles. We co-stained cells infected with wild-type M81 with antibodies specific to CEP170 and to centrin. This analysis revealed that more than two-thirds of

cells that displayed increased centriole numbers had undergone centriole overduplication. This proportion fell to approximately one-third in cells infected with M81/ΔZR, showing that, in these cells, centrosome amplification more frequently results from centrosome accumulation. We attempted to link the observed centrosome overduplication with an alteration in the expression level of proteins involved in the control of centrosome duplication. However, cells infected by M81 or M81/ΔZR expressed the Plk4 protein, a master regulator of centrosome duplication[21], at similar levels (Fig. 6e and Supplementary Fig. 5). Similar results were obtained with immunoblots performed with antibodies specific for SAS-6 and STIL, two other proteins involved in centrosome replication.

**Treatment with EBV particles induces CIN in dividing cells.** The results gathered so far showed that EBV lytic replication increases aneuploidy and centrosome amplification. However, in most infected cell populations, an average of 5% of the cells undergo lytic replication[22]. This subpopulation cannot account for the much higher aneuploidy and CIN rate observed in cells infected with replicating viruses. However, cells undergoing virus replication produce virions that bind to neighbour B cells in the

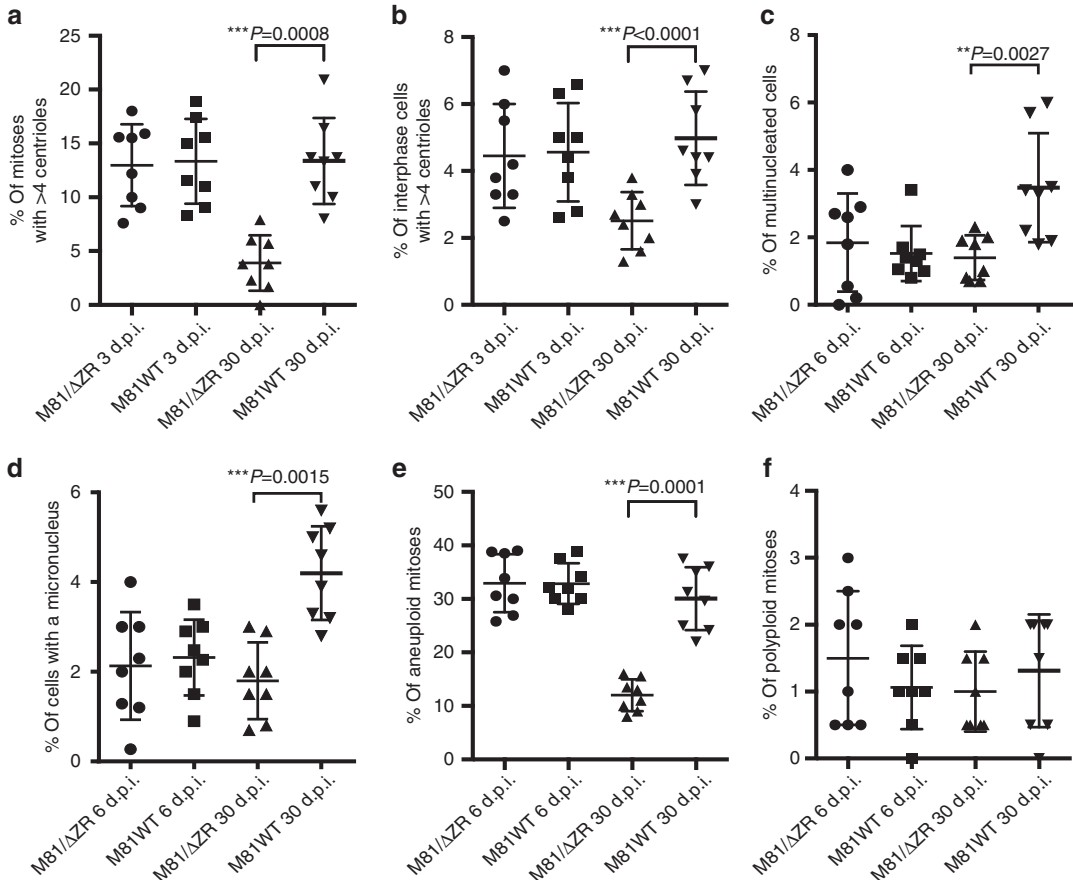

**Figure 2 | Rate of chromosomal instability in cells transformed by wild-type EBV (M81WT) or a replication-defective mutant (M81/ΔZR).** We have analysed eight sample pairs. The cells were analysed at day 3, 6 or 30 post infection. The cells were cytospinned and stained with multiple markers. For each sample, at least 100 mitoses and 500 interphase cells were analysed. Independently, chromosomes were prepared to evaluate the rate of aneuploidy and for each of these samples at least 50 mitoses were analysed. The figure summarizes the frequency of bipolar mitoses organized around more than four centrioles (**a**), of interphase cells with more than four centrioles (**b**), of multinucleated cells (**c**), of cells carrying one or several micronuclei (**d**), of aneuploid mitoses (**e**), of polyploid mitoses (**f**). The graphs include the results of statistically significant paired two-tailed *t*-tests performed on pairs of samples analysed at day 30 post infection (**a**) $P = 0.008$, (**b**) $P < 0.0001$, (**c**) $P = 0.0027$, (**d**) $P = 0.0015$, (**e**) $P = 0.0001$). Error bars represent the mean with s.d. d.p.i., days post infection.

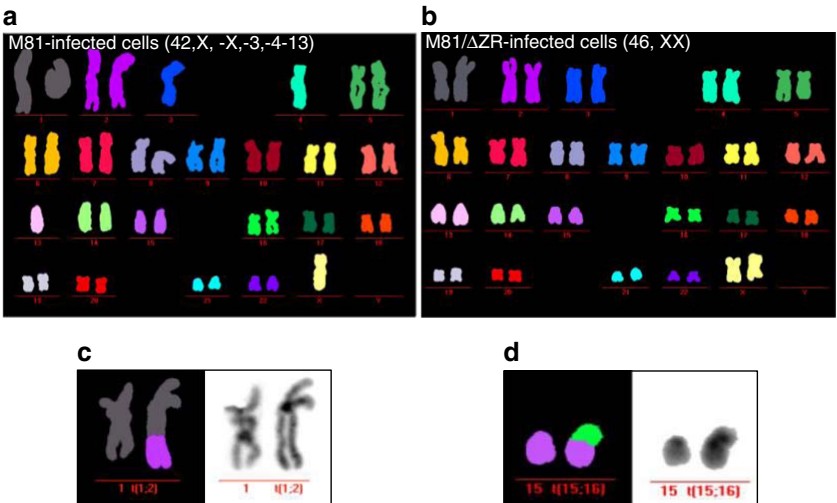

**Figure 3 | B cells transformed by wild-type EBV display a higher CIN rate 4 weeks post infection.** Example of a M-FISH karyotype showing mitoses from a pair of transformed cell lines infected with wild-type EBV (**a**), or with a replication cell-deficient mutant (**b**). (**c,d**) Two translocations are shown, found in two other cell lines transformed by wild-type EBV.

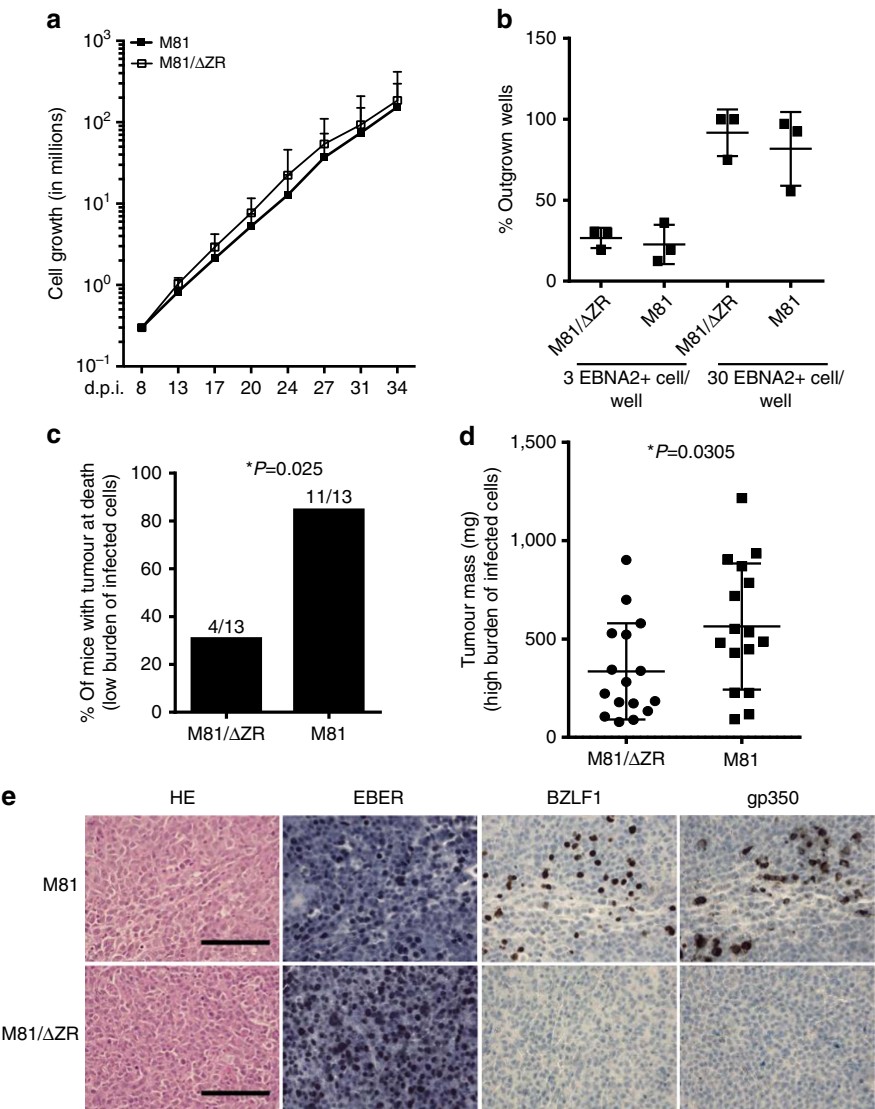

**Figure 4 | B cells infected with wild-type M81 induce tumours with a higher frequency in immunodeficient mice.** B cells were exposed to M81 wild type and to the M81/ΔZR mutant and were injected intraperitoneally to NSG mice or grown *in vitro*. (**a**) The graphs show cell growth of seven independent B cell samples *in vitro* for a period of 34 days. We show the mean value with standard deviation. (**b**) Three of the samples described in **a** were seeded in 96-well cluster plates coated with feeder cells at a concentration of 3 or 30 EBNA2-positive cells per well. The dot plot shows the percentage of outgrown wells taken as a marker of transformation. (**c**) The graph shows the incidence of tumours in 26 immunocompromised mice after injection of $4 \times 10^4$-infected B cells. The results obtained with wild-type M81 and M81/ΔZR were assessed by an exact Mantel–Haenszel test with strata to take into account the variability due to the use of three infected primary B cell samples in this experiment ($P = 0.025$). (**d**) The dot plot shows the tumour mass in 16 animal pairs that developed a tumour after injection of $4 \times 10^5$ B cells infected with M81 or M81/ΔZR. The results are analysed by an unpaired two-tailed *t*-test ($P = 0.0305$). (**e**) Histological stainings showing the morphology of tumours that developed after injection of EBV-infected cells in immunocompromised mice (haematoxylin and eosin stain), the expression pattern of the EBER noncoding RNAs, as well as of the BZLF1 and gp350 proteins. We show one example of a tumour that developed after infection with the wild-type virus or after infection with the M81/ΔZR mutant. Scale bar, 100 μm. Error bars represent the mean with s.d. d.p.i., days post infection.

infected B cell population[22]. We tested whether these bound particles could generate the genetic abnormalities observed in B cells transformed with wild-type EBV by treating LCLs generated with the M81/ΔZR mutant with virus-like particles (VLP) that are devoid of viral DNA and cannot establish a chronic infection[23,24]. The cells were exposed for 3 days to purified particles to exclude contamination with soluble factors from the supernatant. We tested VLPs derived from both B95-8 or from M81. This treatment led to at least a doubling in the frequency of centrosome amplification and aneuploidy, after either type of VLP infection (Fig. 7a–c, Supplementary Fig. 6). Importantly, this

property was not shared by VLPs that are not able to fuse with their targets because they are devoid of the gp110 protein that is required for cell entry[25]. As we found no difference between VLPs derived from either B95-8 or M81, we concentrated on M81 VLPs that can be produced at much higher levels. We added M81 VLPs to B cells expanded by the CD40L system in the presence of IL4 and obtained very similar results in these EBV-negative cells (Fig. 7d–f). We also treated PWM-stimulated B cells, RPE-1 and HeLa cells with VLPs under the same conditions and also observed an increase in the percentage of cells carrying abnormal centrosome numbers (Fig. 7g–j, Supplementary Fig. 7a–h).

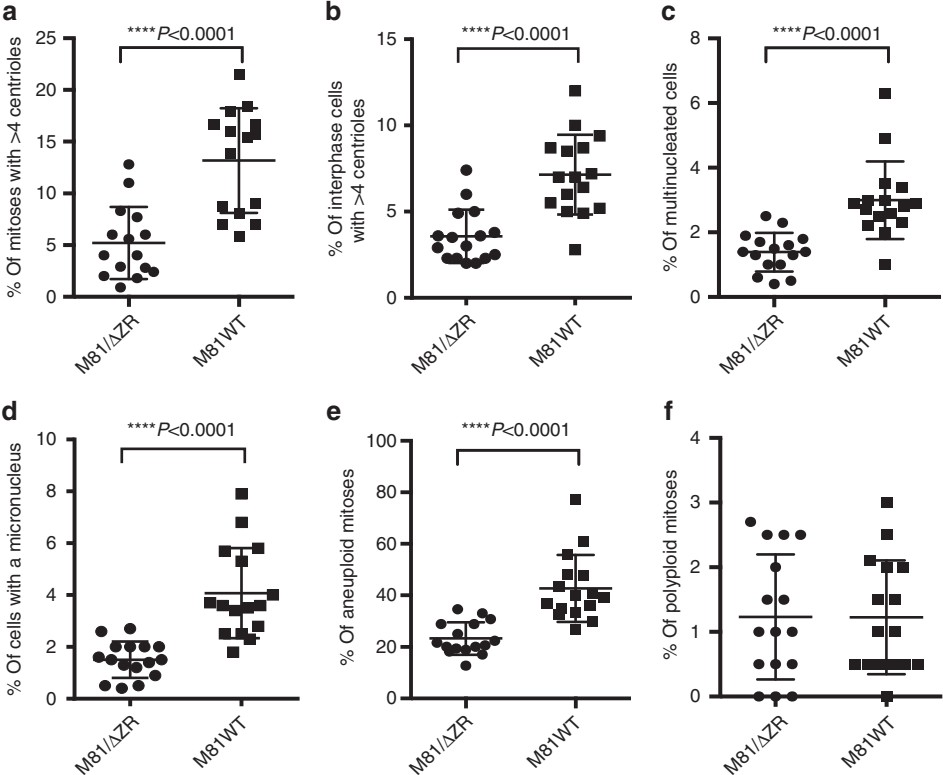

**Figure 5 | Lymphoid tumours generated with wild-type M81 exhibit a higher degree of CIN.** 32 NSG mice were injected with $4 \times 10^5$ B cells infected with wild-type M81 or the M81/ΔZR mutant. The dot plots summarize the frequency of bipolar mitoses organized around more than four centrioles (**a**), of interphase cells with more than four centrioles (**b**), of multinucleated cells (**c**), of cells carrying one or several micronuclei (**d**), of aneuploid mitoses (**e**), of polyploid mitoses (**f**). For each except in two samples, at least 100 mitoses and 500 interphase cells were analysed. The results were subjected to an unpaired two-tailed $t$-test (**a**–**e**) $P < 0.0001$. Error bars represent the mean with s.d. See also Fig. 2 for a comparison with the results of *in vitro* infections.

Similar results were obtained with RPE-1 cells stably transfected with a GFP-centrin-1 fusion protein (Supplementary Fig. 7i–m). We infected the epithelial cell lines RPE-1 and HeLa with wild-type M81 and stained them for expression of the EBV-specific EBER noncoding RNAs (Supplementary Fig. 7n–q). This staining showed that M81 infects between 1 and 1.5% of these cells. These results suggest that EBV can interact with the cell division machinery of cells lines without necessarily being able to establish a chronic infection. M81 VLP treatment of RPE-1 cells also doubled the rate of cells present in cytokinesis, suggesting that this process is delayed by the treatment (Supplementary Fig. 7r). We addressed this issue by exposing HeLa cells stably transfected with mEGFP-α-tubulin and H2B-mCherry fusion proteins to EBV VLPs and performed life cell imaging (Supplementary Fig. 8). Although the average mitotic time was not influenced by the treatment, cytokinesis took significantly longer in cells treated with VLPs or wild-type virus (Supplementary Movies 1 and 2, Supplementary Fig. 8).

**BNRF1 induces centrosome overduplication and aneuploidy.** We then expressed any of 66 EBV proteins in the 293 cell line to assess their contribution to CIN. We found that transfection of BNRF1, a protein that strongly potentiates the efficiency of EBV infection[26], doubled the frequency of centriole amplification and nearly tripled the frequency of multipolar mitoses, relative to mock-transfected cells (Supplementary Fig. 9). Staining for CEP170 revealed that transfection with BNRF1 did not increase the frequency of cells carrying more than two CEP170-positive

centrioles, suggesting that this viral protein causes centriole overduplication (Supplementary Fig. 9i). We monitored BNRF1 expression in primary B cells exposed to EBV. This protein was clearly detectable in the infected B cells during the first 5 days after infection (Supplementary Fig. 10). This observation suggests that the levels of BNRF1 protein are not reduced by cell division in the first days post infection and fits with the observation that EBV-infected B cells do not initiate cell division before 3 days after infection[27]. They also fit with the kinetic of centrosome amplification that was visible at day 3 post infection, at which time point BNRF1 is still available to infected cells. We then repeated the aforementioned superinfection experiments with wild-type B95-8 or with a defective B95-8 mutant that lacks BNRF1 (ref. 26). Although exposure of LCLs generated with the replication-deficient M81/ΔZR mutant to a recombinant B95-8 EBV devoid of the BNRF1 gene (B95-8/ΔBNRF1) did not increase centriole numbers in these cells (Fig. 8a–c), exposure to wild-type viruses or to B95-8/ΔBNRF1 viruses trans-complemented with a BNRF1 expression plasmid did. We also infected primary B cells with BNRF1 knockout viruses derived from either B95-8 or M81 wild-type viruses and found that the growing cells showed a striking 5- to 10-fold reduction in the average frequency of centriole amplification and multipolar mitoses, relative to cells infected with wild-type viruses (Fig. 8d–f,h–j). Complementation of these defective BNRF1 knockout viruses with the BNRF1 protein to reconstitute a wild-type virus restored the abnormalities. Similar, though less pronounced, effects were visible on the rate of aneuploidy. Primary B cells infected with either B95-8/ΔBNRF1 or M81/

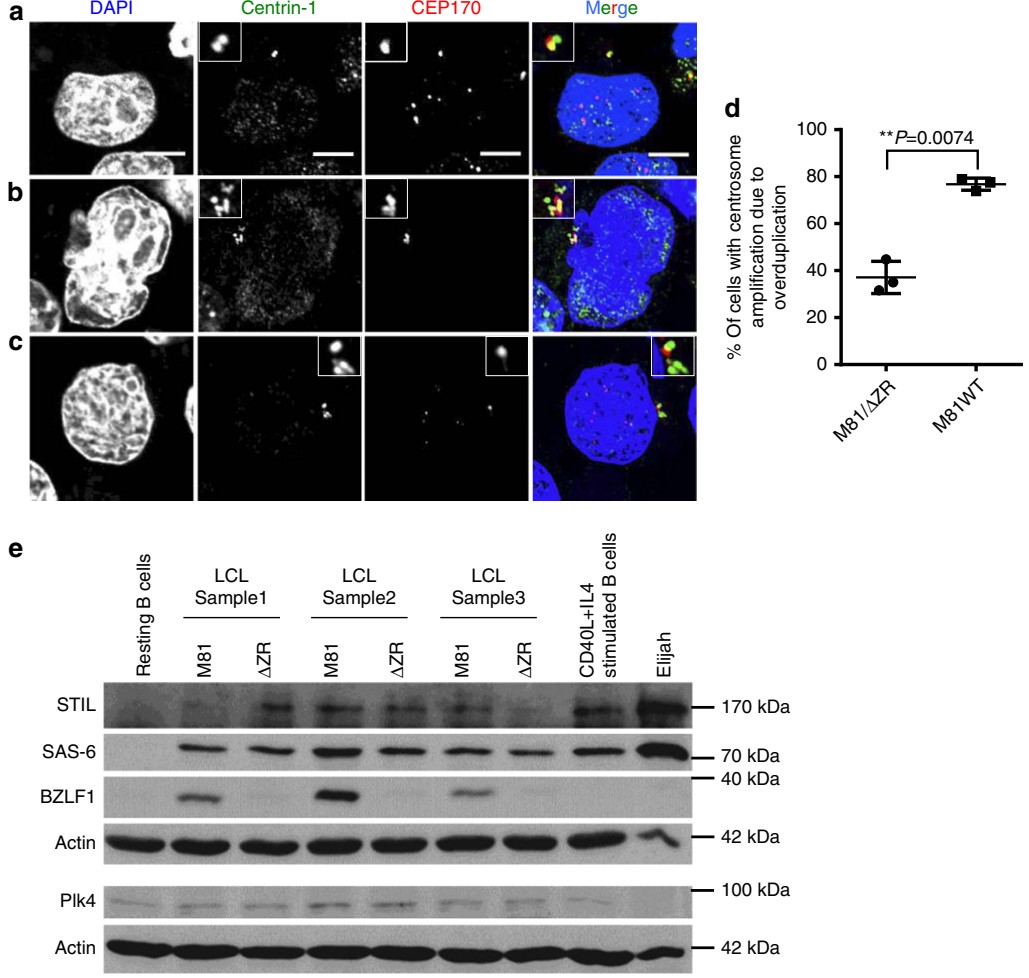

**Figure 6 | Centrosome amplification in wild-type EBV-infected cells mainly results from overduplication.** LCLs infected with wild-type virus were co-stained with antibodies specific to centrin-1 and CEP170, 2 proteins that localize to the centrosome, and counter-stained with DAPI. Scale bar, 5 μm. (**a**) This infected cell shows two centrin-positive centrioles (green) but only one centriole (red) expresses CEP170. (**b**) Infected bi-nucleated cell with centrosome accumulation showing staining for CEP170 in approximately 50% of the centrioles. (**c**) Infected cell showing centrosome overduplication with only one CEP170-positive centriole and at least four centrin-positive centrioles. (**d**) This graph shows the proportion of the cells with centrosome amplification that arose through overduplication in cells infected with wild-type M81 or with M81/ΔZR. The analysis was performed on three blood samples and the results were subjected to a paired two-tailed *t*-test (*P* = 0.0074). Error bars represent the mean with s.d. (**e**) Immunoblots performed on three pairs of LCLs infected with either M81 or M81/ΔZR with antibodies specific to Plk4, Sas-6, STIL, actin or BZLF1. Non-infected resting B cells, B cells stimulated with CD40L and IL4, and the Burkitt's lymphoma cell line Elijah served as controls. We also included a western blot performed on U2OS, RPE-1 and HeLa cells with the same antibodies as controls in Supplementary Fig. 5. Please also see Supplementary Figs 11 and 12 for the uncropped full blots.

ΔBNRF1 virus displayed a 2.5- to 3.5-time reduction in the rate of aneuploidy relative to infection with wild-type or BNRF1-complemented viruses (Fig. 8g,k).

**The BNRF1 protein localizes to the centrosomal fractions.** In an attempt to gain some insights into the mechanisms that underlie BNRF1's ability to induce centrosome amplification, we generated stable 293 cell lines that express BNRF1 under the control of a tetracyclin-responsive promoter. This allows immediate induction in more than 90% of the cells. After exclusion of the nucleus, the cellular organelles were separated on a sucrose gradient. Western blot with antibodies specific to γ-tubulin and centrin-2 allowed identification of the gradient fractions that contained the centrosome (Fig. 9). Immunoblot with a BNRF1-specific antibody revealed that BNRF1 is exclusively located in the centrosome fractions. We also stained the sequential sucrose fractions with antibodies specific to

nucleophosmine (NPM1) and to human poly (ADP-ribose) polymerase 1 (PARP1). As previously described in the literature, both proteins also sedimented in the centrosomal fractions[28,29]. The expression levels of both cellular proteins was similar in the presence or absence of BNRF1, although the shorter form of PARP1 generated by caspase cleavage was overrepresented in cells expressing the viral protein.

## Discussion

Lytic replication has long been recognized as a major risk factor for the development of EBV-positive nasopharyngeal carcinoma. Moreover, cells that undergo EBV lytic replication also activate cellular cancer-associated pathways[30]. However, herpesvirus lytic replication typically leads to cell death, rendering the link between replicating cells and cancer development not obvious[31,32]. Our results remove these conceptual difficulties by showing that the virions themselves conferred the risk induced by lytic replication to non-replicating cells. The EBV particles

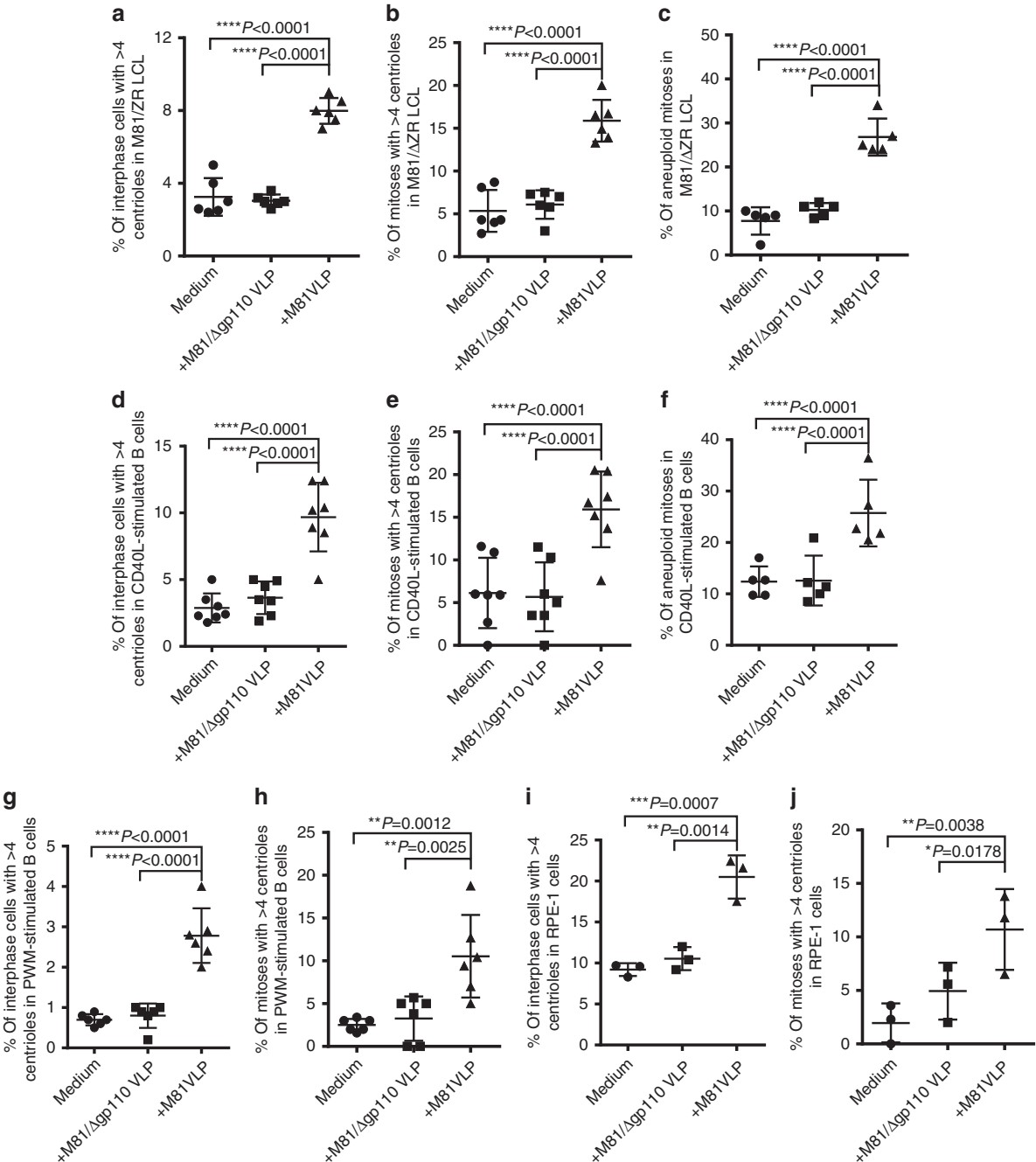

**Figure 7 | Exposure of B cells or RPE-1 cells to virus-like particles leads to centriole amplification.** The different cell populations were treated with M81 virus-like particles (VLP), with virus-like particles that cannot fuse with target cells (Δgp110 VLP) or with medium. The analysis was performed 3 days post infection. For each sample, at least 100 mitoses and 500 interphase cells were examined. The dot plots show the frequency of interphase cells with centriole amplification, bipolar mitoses with an increased number of centrosomes or of aneuploid mitoses in (**a–c**) six B-cell samples transformed by the M81/ΔZR mutant; (**d–f**) seven B-cell samples stimulated with IL4 and CD40-L. We also quantified the percentage of interphase cells or of mitoses that displayed more than four centrioles in six B-cell samples stimulated with pokeweed mitogen (**g** and **h**), and in RPE-1 cells subjected to three independent infections (**i** and **j**). The P values give the results of global mixed-linear model analyses with random effect and we show Bonferroni-adjusted pairwise comparisons (**a–g**) $P < 0.0001$, (**h**) $P = 0.0012$ and $0.0025$, (**i**) $P = 0.0007$ and $0.0014$, (**j**) $P = 0.0038$ and $0.0178$). Error bars represent the mean with s.d.

induced the development of CIN in a few days after infection. This property is substantially lost in a virus that lacks BNRF1 but reappears upon reintroduction of the protein through complementation. Furthermore, transfection of BNRF1 causes centrosome overduplication, an abnormality that facilitates the occurrence of multipolar mitoses and of bipolar mitoses with clustered multiple centrosomes[33]. An increased number of centrosomes is a strong risk factor for defective chromosome attachment, in particular merotelic attachments that foster the occurrence of chromosome lagging and non-aligned chromosomes[33]. Non-aligned chromosomes can lead to micronuclei formation, an anomaly that we observed in EBV-infected cells

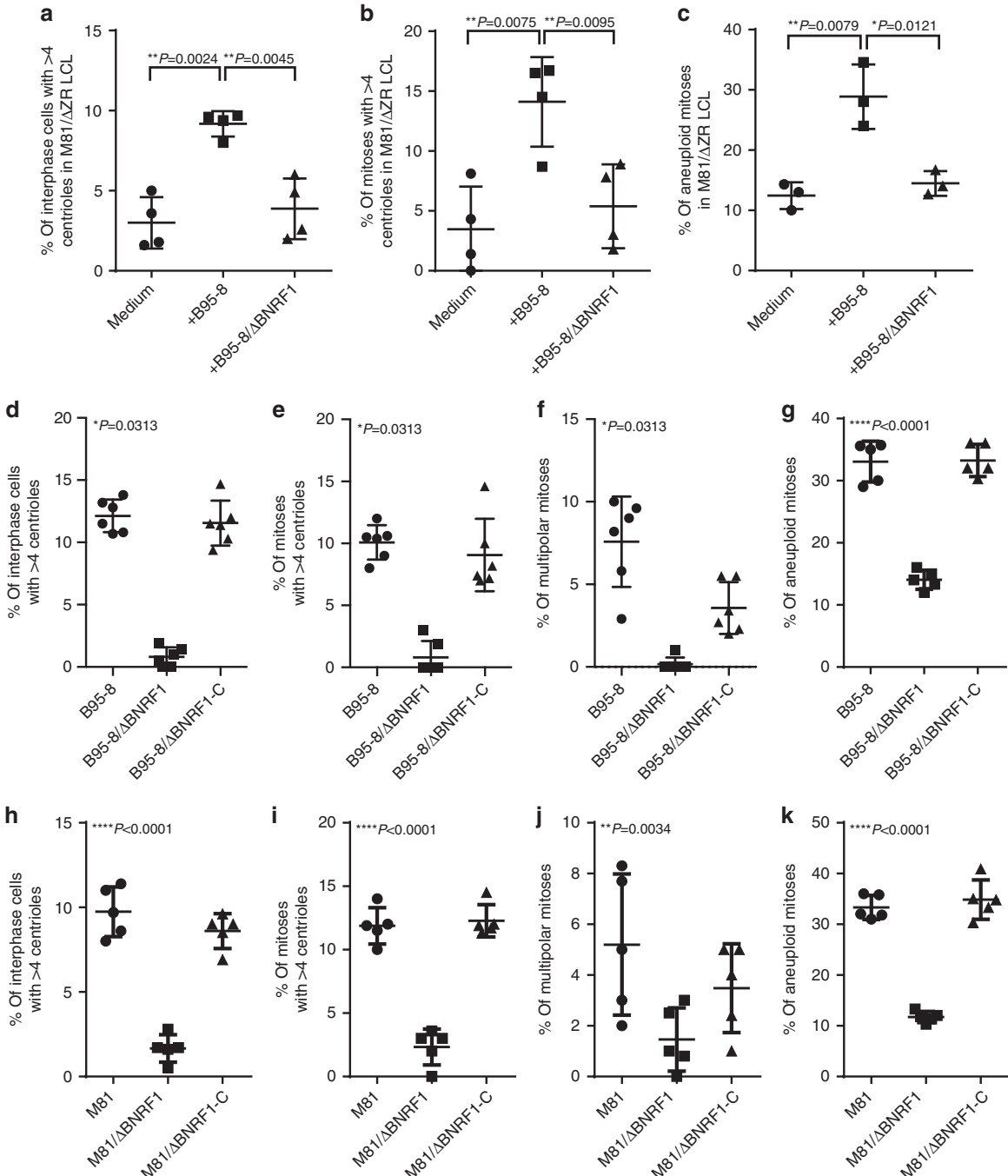

**Figure 8 | B cells infected with Epstein–Barr viruses that lack BNRF1 show a markedly reduced rate of centrosome amplification and aneuploidy.**
(**a**–**c**) Rate of centrosomal amplification and aneuploidy in four independent B-cell samples transformed by M81/ΔZR viruses and exposed to wild-type B95-8 or B95-8/ΔBNRF1 virus. The analysis was performed 3 days after infection. These dot plots summarize the frequency of interphase cells with more than four centrioles (**a**), of bipolar mitoses organized around more than four centrioles (**b**), of aneuploid mitoses (**c**). The results were evaluated with a paired *t*-test. (**d**–**g**) LCLs from at least five independent blood samples were generated with wild-type B95-8, a B95-8/ΔBNRF1 knockout virus or with a B95-8/ΔBNRF1 virus complemented with BNRF1 (ΔBNRF1-C). The dot plots show the frequency of interphase cells harbouring an increased number of centrioles (**d**), of bipolar mitoses organized around more than four centrioles (**e**), of multipolar mitoses (**f**) and of aneuploid mitoses (**g**). (**h**–**k**) Same experiments as **d**–**g**, but performed with a BNRF1 knockout virus constructed on the basis of M81. For each sample, at least 100 mitoses and 500 interphase cells were examined. For **a**–**c**, we give the results of paired two-tailed *t*-tests (**a**) $P = 0.0024$ and 0.0045, (**b**) $P = 0.0075$ and 0.0095, (**c**) $P = 0.0079$ and 0.0121). For **d**–**f**, we applied an exact Wilcoxon signed-rank test and for **g**–**k**, we applied the results of global mixed-linear model analyses with random effect to compare the abnormality rate of B cells infected with ΔBNRF1 mutant with those of B cells infected with wild-type or complemented virus. The *P* values give the results of global mixed-linear model analyses with random effect (**d**–**f**) $P = 0.0313$, (**g**–**i**) $P < 0.0001$, (**j**) $P = 0.0034$, (**k**) $P < 0.0001$). Error bars represent the mean with s.d.

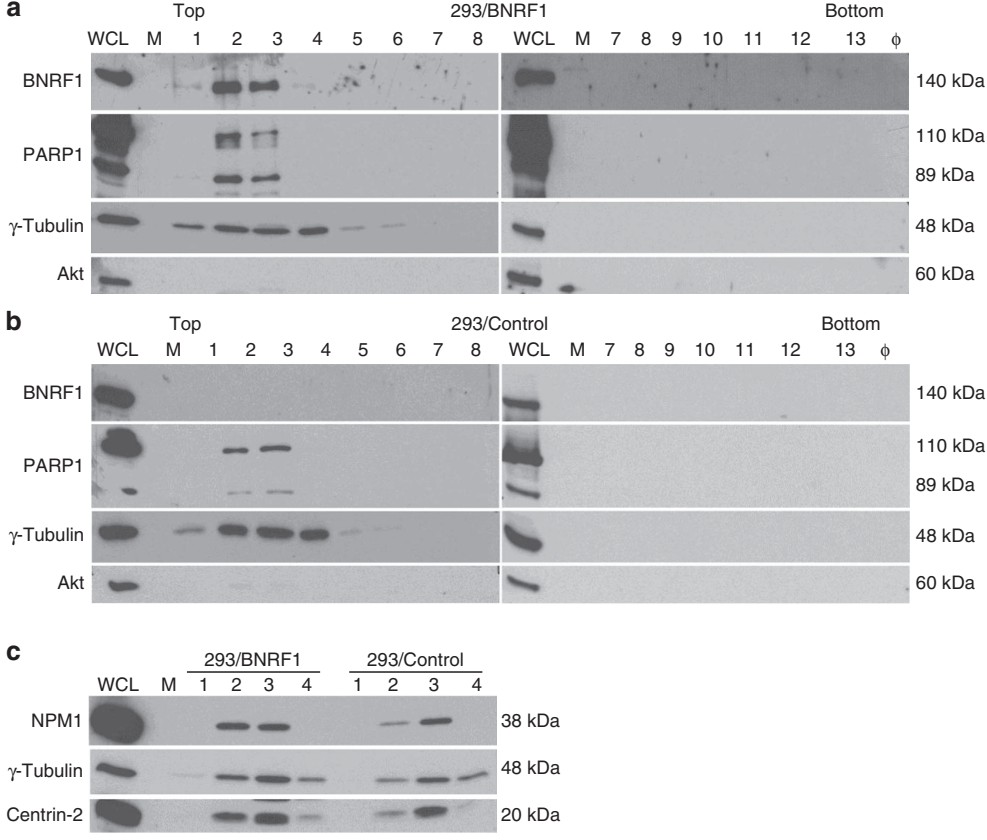

**Figure 9 | BNRF1 is enriched in the centrosomal fraction.** A total 293 cells were subjected to BNRF1 overexpression. Cellular organelles were separated on a sucrose gradient after exclusion of the nuclei. We immunostained the consecutive fractions collected from this gradient with an antibody specific for γ-tubulin to identify the centrosomal proteins and with an antibody specific to BNRF1. We also stained the extracts with antibodies specific to PARP1, a protein that localizes to the centrosome. The antibody specific to PARP1 identifies a full size protein as well as a smaller form of the protein generated by caspase 3 cleavage. Finally, we stained the blots with an antibody specific to Akt to detect contaminations from free cytoplasmic proteins. The latter staining was performed to ensure that the gradient had not been contaminated with free cytoplasmic proteins. Non-purified whole-cell extracts (WCL) of cells with BNRF1 overexpression were included as a positive control and φ indicates wells without samples. (**a**) Fractions collected from cells that expressed BNRF1. (**b**) Fractions collected from cells transfected with an empty control plasmid. (**c**) The fractions containing the centrosomal proteins and described in **a** and **b** were immunoblotted with antibodies specific to NPM1, γ-tubulin and centrin-2. We performed this experiment twice and obtained the same results. We show here one representative result. Please also see Supplementary Fig. 11 for the uncropped full blots.

and can foster extensive genomic rearrangements[34,35]. Lagging chromosomes increase the average time of mitosis and facilitate the development of aneuploidy or of chromosome deletion, for example, if they become blocked in the furrow during cytokinesis[36]. Together with non-aligned chromosomes, they can also cause mitotic slippage that leads to polyploidy with cells carrying either an enlarged nucleus or multiple nuclei[33]. Thus, an aberrant centrosome multiplication induced by BNRF1 could in principle explain all abnormalities identified shortly after EBV infection and upon inception of lytic replication in infected cells.

However, although deletion of BNRF1 in the virion markedly reduced the aneuploidy rate, some residual aneuploidy persisted in the infected cells. This could suggest that the influence of VLPs on the mitotic machinery is not limited to BNRF1 and the centrosome. In this line, we noticed cytokinesis defects in HeLa cells exposed to EBV and multinucleated cells in EBV-infected LCLs. However, the rate of aneuploidy in cells infected with the BNRF1 knockout virus and in CD40L-stimulated cells was nearly identical, thereby suggesting that B cells growing in culture for several weeks always display a low level of aneuploidy, although previous studies reported that non-B diploid cells passaged *in vitro* display approximately 1% of mitoses with figures of

missegregation[37]. It is also important to note that studies with the BNRF1 knockout virions were conducted with very large amounts of viruses, as these viruses have a markedly reduced ability to infect B cells[26]. These conditions might reveal the properties of proteins other than BNRF1 present in EBV infectious particles that usually do not or only minimally interfere with the mitotic machinery.

We also noted that the rate of aneuploidy was higher than the rate of centrosome accumulation in infected cells. This can be partly explained by the methodology we used to unequivocally identify cells with centrosome accumulation. Indeed, we counted non-mitotic cells as abnormal only if they carry more than four centrioles, as normal cells that went through centrosome replication carry four centrioles. However, cells with three or four centrioles could be abnormal cells that had not gone yet through centrosome replication. Moreover, the identification of mitoses organized around an abnormal number of centrioles is only possible if these centrioles are well separated after staining and it is not always the case. Thus, we most certainly underestimated the frequency of centrosome aberrations. Finally, it is theoretically possible that cells with centrosome amplification give rise to daughter cells that received an abnormal number of chromosomes but only one centrosome upon cell division.

We observed translocations or chromosome deletions at a very low rate and no clonal chromosomal abnormality shortly after infection. In contrast, two out of four cell lines infected with wild-type M81 carried a clonal abnormality (one clonal chromosome loss and one clonal translocation) 6 weeks after infection, as did the only LCL infected with B95-8 we investigated (one clonal translocation). This suggests that the cell populations that carried the clonal abnormality were selected out, presumably because these genetic rearrangements confer a growth advantage. Thus, even if translocation is a relatively rare event after EBV infection, its effects can become dominant within the infected population.

Because the CIN induced by the initial contact with viral particles from non-replicating strains subsides over time although these cells continuously express the latent proteins, the effects of EBV infection on the mitotic machinery are probably independent of them[1]. However, it is very likely that the EBV latent proteins modulate the effects of the virions on EBV-infected cells. For example, EBNA3C has been shown to overcome the mitotic checkpoint that should be activated in case of chromosomal instability[38]. Furthermore, EBV latent genes have been found to promote genomic instability and could amplify the effects induced by the virions[39–41].

The central role of the centrosome in organizing the mitotic process explains its frequent deregulation in various types of cancer, including those induced by the transforming papillomavirus strains[42]. The E7 protein expressed from the viral genome in infected cells induces centrosome overduplication[42]. Although not formally demonstrated for EBV, herpesviruses usually use the cytoplasmic microtubule network and its organizer, the centrosome, to reach the nucleus[43,44]. Therefore, herpes viral particles will have an increased probability of interacting with the centrosome. BNRF1 is a component of the tegument whose absence impairs the transfer of the virus particle containing the nucleocapsid from the endosome to the nucleus[26]. Migration of the virus along the tubulin network would offer an opportunity of interaction between BNRF1 and the centrosome. Alternatively, we find that BNRF1 can localize to the centrosomal compartment upon overexpression. It is therefore conceivable, although currently entirely speculative, that BNRF1 brought into the cell through infection but dissociated from the tegument during the course of infection could reach the centrosome on its own.

The mechanisms that underlie the ability of BNRF1 to induce centrosome accumulation remain unclear. We could not find any differences in the expression levels of the centrosome-regulating proteins STIL, SAS-6 or Plk4 between B cells infected by replicating or non-replicating viruses. BNRF1 has been found to bind to DAAX proteins in the nucleus[45]. DAXX can also transiently co-localize to the centromere of chromosomes after heat-shock treatment[46]. Immunoprecipitation experiments using an antibody against BNRF1 coupled with protein sequencing by mass spectrometry also identified NPM1 and PARP1, two proteins present in the centrosome, as binding partners of BNRF1 (ref. 45). However, the interaction with these proteins, if real, must be weak as their interactions with BNRF1 could not be confirmed in subsequent experiments. We found that BNRF1 expression increases the production of the truncated PARP1 form generated by caspase cleavage within the centrosome. Whether or not the increase of this particular PARP1 form reflects the effects of BNRF1 on the centrosome is currently unclear.

We show that replication-competent viruses induce tumours in immuno-compromized mice with a higher frequency than replication-defective ones. Furthermore, tumours induced by wild-type viruses displayed stronger CIN and aneuploidy rates. This suggests that wild-type viruses are more transforming in vivo because they induce more CIN and aneuploidy. However, this effect was not visible in vitro. The impact of centrosome

overduplication in tumorigenesis is difficult to evaluate in experimental models and some authors have previously suggested that it can only be revealed under in vivo conditions[47]. Our observations are concordant with similar experiments previously performed in humanized mice that showed that the wild-type B95-8 strain is more transforming than a mutant thereof that lacks the Z transactivator, although macroscopic tumours rarely develop in these animals[48]. Moreover, humanized mice have an immune system that interferes with infected cells. As B95-8 induces a very limited and abortive lytic replication, it is unlikely that the effects observed in this work are due to the production of virions.

The present work largely explains the high degree of aneuploidy that was previously reported in B cells infected in vitro[49] or in infected cells from healthy individuals propagated in immunocompromised mice[50], as well as in several types of EBV-induced lymphomas[51,52] and carcinomas[53–55]. In particular, PTLD frequently show signs of lytic replication and aneuploidy, and aneuploid tumours have a poorer response to treatment[51,56–58].

Our work suggests that every EBV infection, including a primary infection, increases the risk of CIN and aneuploidy, a risk factor for cancer development and that this risk will increase proportionally to the frequency of contact with EBV virions. Thus, patients infected with viruses that strongly replicate or whose immune system cannot control virus multiplication such as patients with immunodeficiencies are likely to be at higher risk[56,59]. Our observations also have consequences for vaccination strategies as VLPs have previously been proposed as vaccines[23,24,60]. Deletion of BNRF1 from these viruses, or keeping the virus from reaching the centrosome, should circumvent this potential problem.

We also found that the effects of EBV virions extend to EBV-negative cells. Because the virus does not have to induce a stable infection to exert its effects on the centrosome, the range of cell lineages in which the virus could increase the risk of CIN could theoretically extend beyond the classical EBV targets. Thus, EBV could be a risk factor for tumour development without being present in the resulting tumour. It is interesting to note that individuals who underwent an episode of infectious mononucleosis after primary EBV infection are not only at increased risk of EBV-associated Hodgkiń́s disease, but also of non-Hodgkin lymphomas during the first year after infection[61–63]. The latter tumours carry the EBV genome only in 5% of the cases[64]. It would be important to perform epidemiological studies that correlate clinical and biological markers of EBV lytic replication with general cancer risk.

## Methods

**Ethics statement.** All human primary B cells used in this study were isolated from anonymous buffy coats purchased from the blood bank of the University of Heidelberg. No ethical approval is required. All animal experiments were performed in strict accordance with German animal protection law (TierSchG) and were approved by the federal veterinary office at the Regierungspräsidium Karlsruhe, Germany (Approval number G156-12). The mice were housed in the class II containment laboratories of the German Cancer Research Centre and handled in accordance with good animal practice with the aim of minimizing animal suffering and reducing mice usage as defined by Federation of European Laboratory Animal Science Associations (FELASA) and the Society for Laboratory Animal Science (GV-SOLAS).

**Cell lines and primary cells and viruses.** The 293 cell line is a neuro-endocrine cell line obtained by transformation of embryonic epithelial kidney cells with adenovirus (ATCC: CRL-1573). HeLa is a human cervix adenocarcinoma cell line (ATCC: CLL-2) that is infected with papillomavirus type 18. HeLa Kyoto mEGFP-α-tubulin/H2B-mCherry cell line is a derivate thereof that stably expresses the mEGFP-α-tubulin and H2B-mCherry protein fusions[65]. RPE-1 is a human epithelial cell line immortalized with hTERT (ATCC: CRL-4000). RPE-1/centrin-1-GFP is a cell line that constitutively expresses a centrin-1-GFP fusion protein[66].

U2OS is a cell line derived from a moderately differentiated sarcoma of the tibia (ATCC: HTB-96). All the cell lines in this study were tested and found to be free of mycoplasma contamination. Peripheral blood mononuclear cells from buffy coats purchased from the blood bank in Heidelberg were purified on a Ficoll cushion and CD19-positive primary B-lymphocytes were isolated using M-450 CD19 (Pan B) Dynabeads (Dynal) and were detached using Detachabead (Dynal). WI38 are primary human lung embryonic fibroblasts (ATCC: CCL-75). All the cells were routinely cultured in RPMI-1640 medium (Invitrogen) supplemented with 10% fetal bovine serum (FBS)(Biochrom), and primary B cells were supplemented with 20% FBS until the establishment of LCLs. HeLa Kyoto mEGFP-α-tubulin/H2B-mCherry cells were supplemented with $0.5 \mu g\,ml^{-1}$ puromycin and $500 \mu g\,ml^{-1}$ G418. The EBV producer cells used in this study (M81, M81/ΔZR, B95-8, B95-8/ΔBNRF1, B95-8/ΔBFLF1ΔBFRF1ΔBBRF1ΔBALF4 (VLP with gp110 deletion), B95-8/ΔBFLF1ΔBFRF1ΔBBRF1 (VLP)) have previously been described and were established by stable transfection of EBV-BACs into 293 cells supplemented with $100 \mu g\,ml^{-1}$ hygromycin[8,22,24,26]. The VLP-producing mutants and the ΔBNRF1 mutant were also available on the basis of the M81 strain. They were constructed exactly as their B95-8 homologues. M81/ΔZR lacks the BZLF1 and BRLF1 transactivators that initiate lytic replication and therefore it is unable to replicate, B95-8/ΔBNRF1 and M81/ΔBNRF1 lack the BNRF1 tegument protein.

**Plasmids.** The BZLF1 (p509), BALF4 (pRA) and BNRF1 (B056) expression plasmids were previously described[26]. We screened a library of 66 EBV proteins driven from a CMV promoter[67]. An expression plasmid that encodes a cytoplasmic-truncated version of rat CD2 (B673) was constructed in pcDNA3.1. We also cloned the BNRF1 gene into a tetracycline-inducible plasmid, containing a minimal CMV promoter controlled by TetO operator, a tetracycline transactivator protein (Tet-On) driven by CAG promoter, the origin of plasmid replication derived from B95-8 strain, and a puromycin resistance cassette driven by a SV40 promoter (B1439)[68]. The parental vector without insert served as a negative control.

**Transfections.** All the transfection experiments were performed with the liposome-based transfectant Metafectene (Biontex) following the manufacturer's instruction.

**Virus production.** The 293 cells stably transfected with recombinant EBV-BACs were transfected with expression plasmids encoding BZLF1 (p509) and BALF4 (pRA) to induce lytic replication, except for the production of VLPs that lack gp110 in which case only the BZLF1-encoding plasmid was transfected. Transfection of a plasmid that encodes the BNRF1 protein (B056) in a producer cell line that stably carries the ΔBNRF1 virus led to trans-complementation as described previously[26]. Three days after transfection, virus supernatants were collected and filtered through a $0.4 \mu m$ filter.

**B-cell stimulation with mitogens or CD40-ligand.** Freshly isolated CD19+ primary B cells were cultured with $15 \mu g\,ml^{-1}$ of PWM (L9379, Sigma-Aldrich) or cultured on a 90 Gy-γ-irradiated CD40-ligand feeder cell layer in the presence of $25 \,ng\,ml^{-1}$ recombinant human IL4 (PeproTech, Germany). The cells were subjected to cytospins or chromosomal analyses 3 days after the inception of stimulation.

**Giemsa staining.** The cells were treated with $0.075 \mu g\,ml^{-1}$ colchicine (Sigma-Aldrich C3915) for 2 h to induce metaphase arrest and allow the preparation of metaphase spreads. After three washings with phosphate-buffered saline (PBS), the cells were incubated in 75 mM KCl hypotonic buffer for 10 min at 37 °C and fixed in methanol: glacial acetic acid (3:1), dropped onto cold glass slides and stained with 5% Giemsa (Carl Roth GmbH T862.1) in water. Digital images of metaphase were captured using DM2500 (Leica, Wetzlar, Germany) microscope equipped with a DFC300 FX (Leica, Cambridge, UK) camera and subjected to karyotyping. We analysed a minimum of 50 mitoses per sample. The investigator was blinded to the group allocation. The experiments were performed single blinded. Fifteen samples were independently analysed in parallel by Giemsa staining and 24-colour chromosomal painting and yielded very similar results (see below).

**Multiplex fluorescence in situ hybridization.** M-FISH was performed as previously described[69]. Briefly, seven pools of flow-sorted human whole chromosome painting probes were amplified and directly labelled with seven different fluorochromes (DEAC, FITC, Cy3, Cy3.5, Cy5, Cy5.5 and Cy7) using degenerated oligonucleotides and PCR (DOP-PCR). Metaphase chromosomes immobilized on glass slides were denatured in 70% formamide/2xSSC pH 7.0 at 72 °C for 2 min followed by dehydration in increasingly pure ethanol series. The hybridization mixture contained combinatorially labelled painting probes, an excess of unlabelled cot1 DNA, 50% formamide, 2xSSC and 15% dextran sulfate. It was denatured for 7 min at 75 °C, pre-annealed at 37 °C for 20 min and hybridized at 37 °C to the denatured metaphase preparations. After 48 h, the slides were washed in 2xSSC at room temperature three times for 5 min, followed by two

washes in 0.2xSSC/0.2% Tween-20 at 56 °C for 7 min each. Metaphase spreads were counterstained with 4.6-diamidino-2-phenylindole (DAPI) and covered with antifade solution. Metaphase spreads were recorded using a DM RXA epifluorescence microscope (Leica Microsystems, Bensheim, Germany) equipped with a Sensys CCD camera (Photometrics, Tucson, AZ, USA). Camera and microscope are controlled by the Leica Q-FISH software and the images were processed on the basis of the Leica MCK software and presented as multicolor karyograms (Leica Microsystems Imaging solutions, Cambridge, UK). We analysed between 15 and 20 metaphases for each sample.

**Analysis of the mitotic spindle.** The cells were washed three times and resuspended in PBS-3% FBS. The single cell suspension was then loaded on to Shandon cytospin chambers with slides (Thermo Scientifics) and spun at 2,000 r.p.m. for 10 min. The cytospinned cells were air-dried, fixed in pure methanol at −20 °C for 8 min and briefly washed in PBS two times at room temperature for 5 min each. The cells were blocked in PBS-3% bovine serum albumin (BSA) for 30 min, incubated with the first antibody for 1.5 h, washed in PBS three times for 5 min, incubated with a secondary antibody conjugated to Cy-3, Cy-5 or Alexa488 for 1.5 h. The slides were again washed three times in PBS and mounted in ProLong Gold antifade reagent including the DAPI fluorochrome (Life Technologies). In each sample, at least 100 mitoses and 500 interphase cells were examined. The investigator was single blinded for the analysis of the samples. Pictures of stained cells were taken with a camera attached to a DM2500 fluorescence microscope (Leica) or with a confocal microscope (Zeiss LSM700 run on ZEN2009).

**Cell cycle synchronization.** HeLa Kyoto mEGFP-α-tubulin/H2B-mCherry cells (or other cells applied in the study) were treated with 2 mM thymidine for 16 h, released for 8 h and again blocked for 16 h to obtain a double thymidine block.

**Life cell imaging.** We performed life cell imaging on HeLa Kyoto mEGFP-α-tubulin/ H2B-mCherry cells that were treated for 72 h with medium, viruses or virus-like particles. During this treatment, the cells were synchronized in the G1 phase by a double thymidine block. After the second release of the thymidine block, $2.5 × 10^5$ cells per well were seeded in Ibidi μ-slide eight-well plate or Lab-Tek II chambered coverglass (eight chambers). The cells were monitored by a × 20/0.4 air objective on an inverted microscope (Zeiss motorized Observer.Z1) connected to a colour CCD camera AxioCam ICc 3 at 5% $CO_2$ and 37 °C incubator. LED module Colibri.2 with 470 nm for GFP and 590 nm for mCherry were used for fluorochrome excitation. Multipoint images were taken with 3–8 z-stacks to cover a range of 6 to $8 \mu m$ every 5 min for 5–15 h with the cell Zeiss Zen blue software. Maximum intensity projection of the fluorescent channels was performed by ImageJ software to create 8-bit RGB TIFF files and movies.

**B-cell infections and in vitro transformation experiments.** B cells purified from peripheral blood of different healthy donors were exposed to viruses for 2 h at a multiplicity of infection of 20 virus genomes, as defined by qPCR per target cell as described previously[26]. The infected cells were washed once with PBS and plated in cluster plates in RPMI supplemented with 20% FBS. For transformation assays, we first determined the percentage of EBNA2-positive cells within the infected sample using immunostaining 3 days post infection (d.p.i.). Infected cell populations were seeded in 96-U-well plates coated with $10^3$ γ-irradiated WI38 feeder cells at a concentration of 3 or 30 EBNA2-positive cells per well. Non-infected B cells served as a negative control. The outgrowth of lymphoblastoid cell clones (LCLs) was monitored at 30 d.p.i. In parallel, we also monitored cell growth in batch culture by counting the cell numbers in the infected populations twice per week.

**Screening of the EBV library.** The EBV protein expression library[67] was used for transient transfection into 293 cells. To identify the transfected cells, we co-transfected a plasmid encoding a cytoplasmic-truncated rat CD2 that is expressed as a surface marker.

**Transformation experiments in immunocompromised mice.** We isolated human CD19+ B cells from buffy coats and exposed them to M81 or M81/ΔZR in vitro for 2 h at room temperature under constant agitation at a multiplicity of infection sufficient to generate 20% of EBNA2-positive cells[8]. We purchased the NSG mouse strain (NOD.Cg-Prkdc^scid Il2rg^tm1Wjl/SzJ; NSG) from the Jackson Laboratory that established it. This strain is currently maintained in the animal facility of the German Cancer Research Center. The pre-established inclusion criteria in this study were healthy male NSG mice aged between 6 and 10 weeks, the exclusion criteria were death unrelated to the virus infection. The infected cells were collected by centrifugation and washed twice with PBS. A total $2 × 10^5$ or $2 × 10^6$ primary B cells exposed to the virus, equivalent to $4 × 10^4$ and $4 × 10^5$ EBV-infected cells, respectively, were injected intraperitoneally into NSG mice that were of similar age and were randomly grouped in different mouse cages. We used three different buffy coats to infect 26 mice with $4 × 10^4$ EBV-infected cells and five different buffy coats to infect 32 mice with $4 × 10^5$ EBV-infected cells. These

numbers ensured adequate power to detect pre-effect size. The mice were killed at 6 weeks post injection when clinical symptoms appeared (apathy, food refusal, ruffled hair, weight loss, palpable tumour). After careful autopsy, the organs were subjected to macroscopic and microscopic investigation, including haematoxylin and eosin staining and immunohistochemistry. We also generated single cell suspensions from the tumour mass that were cultured overnight in RPMI-20% FBS and used to generate metaphase spreads or cytospinned and subjected to immunofluorescence staining. In all of these experiments, the investigator was single blinded for the investigation of the mice and of the mice samples.

**Immunohistochemistry.** Organs from the killed mice were fixed in 10% formalin and embedded in paraffin. Three micrometre thin sections were prepared and immunostained after antigen retrieval (10 mM sodium citrate, 0.05% Tween 20, pH 6.0; 98 °C for 40 min). Bound antibodies were visualized with the Envision + Dual link system-HRP (Dako). Pictures were taken with a camera attached to a light microscope (Axioplan, Zeiss).

**Western blots.** Proteins were extracted with a standard lysis buffer (150 mM NaCl, 0.5% NP-40, 1% sodium deoxycholate, 0.1% SDS, 5 mM EDTA, 20 mM Tris-HCl pH 7.5, proteinase inhibitor cocktail (Roche)) for 15 min on ice followed by sonication to shear the genomic DNA. Up to 20 μg of proteins denatured in Laemmli buffer for 5 min at 95 °C were separated on SDS-polyacrylamide gels and electroblotted onto a nitrocellulose membrane (Hybond C, Amersham). After pre-incubation of the blot in 3% BSA PBST (PBS with 0.2% Tween 20), the antibody against the target protein was added and incubated at room temperature for 1 h. After extensive washings in PBST, the blot was incubated for 1 h with secondary antibodies. Bound antibodies were revealed using the ECL detection reagent (Pierce). Uncropped scans of all immunoblots are shown in Supplementary Figs 11 and 12.

**Antibodies.** We used primary mouse monoclonal antibodies against α-tubulin (Sigma-Aldrich T5168 1:4,000 for immunostains), γ-tubulin (Sigma-Aldrich T6557 1:5,000 for immunoblots), Plk1 (Santa Cruz sc-17783 1:200 for immunostains), SAS-6 (Santa Cruz sc-81431 1:500 for immunoblots), centrin-1 (Millipore 04-1624 1:100 for immunostains), NPM1 (Zymed 32-5200 1:1,000 for immunoblots), beta actin (Dianova DLN-07273 1:10,000 for immunoblots); rabbit polyclonal antibodies against centrin-2 (Santa Cruz sc-27793-R 1:100 for immunostains and 1:1,000 for immunoblots), CEP170 (Abcam ab72505 1:500 for immunostains), phospho-Histone H3 (PH3, Cell Signaling 9716 1:100 for immunostains), STIL (Bethyl Laboratories A302-442A 1:500 for immunoblots), PARP1 (Cell Signaling 9542S 1:1,000 for immunoblots), Akt (Cell Signaling 1:1,000 for immunoblots); human polyclonal anti centromere (CREST, Antibodies Incorporated 15-235-F 1:5 for immunostains). The mouse monoclonal antibodies against BZLF1 (clone BZ.1 1:1,000 for immunoblots and 1:100 for immunohistochemistry), gp350 (clone OT6 1:1,000 for immunoblots and 1:100 for immunohistochemistry), rat CD2 (clone OX34 1:100 for immunostains) were collected from hybridoma supernatants. Rabbit antiserum against BNRF1 protein (1:10,000 for immunoblots) was produced as described before[26]. The mouse antibody against human Plk4 (0.2 μg ml$^{-1}$ for immunoblots) was raised against a synthetic peptide (amino acid 567–579 of human Plk4)[70]. The secondary antibodies applied for immunofluorescence staining were goat anti-mouse coupled to Alexa488 (Invitrogen A11029 1:300) or Cy3 (Dianova 115-165-146 1:300), anti-rabbit coupled to Alexa488 (Invitrogen A11008 1:300) or Cy3 (Dianova 111-165-144 1:300). Horseradish peroxidase-coupled goat anti-mouse or rabbit antibodies (Promega) were applied as secondary antibodies for western blot analyses (dilution 1:10,000).

**Centrosome isolation.** The 293 cells were stably transfected with the tetracycline-inducible plasmid carrying BNRF1 (B1439) or its control (B484) using puromycin selection (2 μg ml$^{-1}$). Single-cell colonies that displayed an induction rate of at least 90% were selected for further experiments. These cells were induced with 0.025 μg ml$^{-1}$ doxycycline for 1.5 days, then treated with nocodazole (10 μg ml$^{-1}$; Merck Millipore) and cytochalasin B (5 μg ml$^{-1}$; Merck Millipore) for 90 min. The cells were re-suspended in ice-cold PBS and the pellets were collected by centrifugation. After washing with 0.1 × PBS, 8% sucrose, the pellets were lysed by adding 8 ml of lysis buffer (1 mM Tris-HCl pH 8.0, 0.5% Nonidet P-40, 0.5 mM MgCl₂, 0.1% β-mercaptoethanol, 1X proteinase inhibitor cocktail (Roche)) per 15 cm plate, inverted several times and put on ice for 5 min. The lysates were spun at 2,500g for 10 min at 4 °C to pellet down the nuclei, aggregates and intact cells. The clarified supernatants were carefully collected and further filtered through 40 μm cell strainers. The lysates were adjusted to 1X PE (10 mM Pipes, 1 mM ETDA) by using 50X PE buffer (500 mM Pipes, 50 mM ETDA, pH 7.2), incubated with 1 μg ml$^{-1}$ DNaseI on ice for 15 min, loaded onto a 50% (weight/weight) sucrose cushion prepared in gradient buffer (1X PE buffer, 0.1% Nonidet P-40, 0.1% β-mercaptoethanol), spun at 4 °C for 20 min at 12,000 r.p.m. with a SW40Ti rotor without break. After centrifugation, 7 ml of supernatant were visible atop of the cushion. We discarded the first 5 ml of supernatant and collected the remaining 2 ml, together with first the millilitre of sucrose gradient. These combined fractions were well mixed and loaded onto a discontinuous sucrose gradient made from bottom to top of 1 ml 70% (weight/weight) sucrose, 1.5 ml 50% (weight/weight)

sucrose, 2.5 ml 40% (weight/weight) sucrose prepared in gradient buffer. The gradients were spun at 34,000 r.p.m. at 4 °C for 90 min with a SW40Ti rotor without break. After centrifugation, the upper supernatant atop of the sucrose gradient was discarded and the sucrose fractions were collected in 450 μl aliquots from the bottom to the top. The organelles present in each fraction were recovered by mixing 100 μl of each fraction with 1.2 ml 1X PE buffer and centrifuging them at 21,000g at 4 °C for 25 min. The supernatants from each these preparations were carefully removed, the pellets were lysed using SDS sample buffer and subjected to SDS page and western blot analyses.

**Statistical analysis.** All the experiments were planned with a biostatistician (A.K.S.). Power calculations were performed for pairwise comparisons of independent groups of quantitative (normally distributed) data. The effect size of interest in this situation was prespecified to attain or exceed 2 (relevant difference/standard deviation). For an effect size of 2, using group sizes of $n = 5$ guarantees the power for a hypothesis test at the 5% significance level of 79.1%, for $n = 8$ power increases to 96% and reaches 99.7% for $n = 12$. The compared groups had a size larger than 12 in animal experiments or in experiments involving cells that grew in vivo, and more than 5 in the large majority of in vitro experiments. In all the compared groups, the standard deviation is shown as an error bar in the plot. This serves as an estimate of variation. In all the groups, the distribution was normal and the variance was similar in the compared groups.

We applied paired Student's t-tests to the data collected from the infection of multiple primary B cell samples or of LCLs established from the same blood sample with two different types of viruses. The results collected from independent infection experiments of the same cell lines with two different viruses were analysed with a paired Student's t-test. We used a mixed-linear model with random effect for donor to globally analyse the effects of exposure to different viruses or of mock-infection, and used Bonferroni-adjustment for pairwise comparisons. The calculations were performed with SAS 9.3. Infection experiments that included negative results were analysed with a Wilcoxon signed-rank test with calculations performed with R. The results of the animal experiments in which multiple B-cell populations were used for infections were evaluated with an exact Mantel–Haenszel test with strata and the calculations performed with R. The data gathered by life cell imaging over time were, as expected, right-skewed and were log-transformed. They were then subjected to an analysis of variance test performed on SAS 9.3, followed by Bonferroni-adjusted pairwise comparisons.

**Data availability.** All relevant data supporting the findings of this study are available within the article and its Supplementary Information files, or from the corresponding author on request.

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

## Acknowledgements

We are grateful to Helmut Bannert, Helge Lips and Kristine Jaehne for excellent technical assistance. We thank Dr Kurt Reifenberg, Dr Michaela Socher and all members of the DKFZ Laboratory Animal Core Facility for excellent animal welfare and husbandry. Support by Damir Krunic and the DKFZ Light Microscopy facility is gratefully acknowledged.

## Author contributions

I.H. and H.-J.D. designed the study. A.S., M.-H.T., Y.T.S., A.-S.K., K.B., S.F., T.M. and X.L. performed the experiments. A.J., J.M., A.K.-S., R.F., I.H. and H.-J.D. analysed the data. M.-H.T., I.H. and H.-J.D. wrote the manuscript. A.S. and M.-H.T. should be considered equal first authors and are listed alphabetically.

## Additional information

**Competing financial interests:** The authors declare no competing financial interests.

