## [Peer Review File · Nature Communications]

Reviewers' comments:

Reviewer #1 (Remarks to the Author):

The manuscript by Shumilov and colleagues nicely demonstrates that EBV infection can cause chromosomal instability and associated centrosomal abnormalities soon after infection of B cells. Interestingly, this is caused by a viral tegument protein (BNRF1) that was not considered to be one of the main EBV oncoproteins. These findings have important implications for the way we think about viral oncogenesis, and could explain some EBV-negative cancers where abortive infection can contribute to cellular transformation (and potentially hit-and-run long-lasting effects).

Specific comments:

1. Extended data Fig 3: The M81 infected cells showed a recurrent t(6;9) and the ones infected with B95-8 showed a recurrent t(9;15). To me, this suggests early expansion of a single clone in these cultures, and perhaps these translocations should not be considered independent events but rather selection of the clones with a translocation. The legend acknowledges that panels C and D are "cell lines", and in panels A and B no translocations were identified, which I assume are shorter-term cultures. Independent infections of different donors followed by karyotyping should be able to clarify whether in long term cultures translocations are more common for M81 than for M81/ Δ ZR. At the very least, the text could be more clear about whether these translocations were only seen in cell lines, and the time between infection and karyotyping in panels A, B, C and D.

2. Line 130 states: "...either with the B95-8 wild type virus or with virus-like particles (VLP) derived from M81 that are devoid of viral DNA and cannot establish a chronic infection." Is this correct? Why was VLP from M81 compared to B95-8? Also in figure 3 there does not appear to be data for B95-8, so this may be a typo.

3. BNRF1 has been shown to bind DAXX, which is a histone H3 chaperone, and displace ATRX from this complex, a process reported to modify chromatin to promote latent EBV gene expression. ATRX is thought to be involved in chromosomal segregation during mitosis. ATRX mutations have also been reported in different types of cancers. DAXX has other interesting functions, including some in the DNA damage response. It would be useful to include some description (and references) of the known activities of BNRF1 and a brief discussion of how the authors think that BNRF1 could be inducing chromosomal instability.

Reviewer #2 (Remarks to the Author):

The manuscript by Shumilov et al. describes an impressive body of varied research that leads the authors to two conclusions: The BNRF1 gene product of EBV induces amplification of centrosomes when it enters cells and this function contributes to tumorigenesis. The

authors suggest a corollary to their second conclusion: That EBV could potentially contribute to tumorigenesis without being retained in the evolving tumor cells. The authors previously have studied this viral gene and found that it is needed for virions to move, upon cellular entry, their nucleocapsid into the nucleus. How a viral protein with this function leads to the detected amplification of centrosomes remains a mystery.

The BNRF1 gene product is not synthesized in newly infected cells but is brought into the infected cell in the tegument of the virion. One uncertainty the authors need to address is the timing with which the amplification proceeds in newly infected cells. They monitor its increase during the first six days following infection and show its decline 30 days after infection. Given that the BNRF1 protein will only be lost as the infection proceeds, it is imperative for the authors to show that the detected amplification of centrosomes reflects the levels of expression of the BNRF1 protein in the infected cells temporally or their first conclusion is untenable. For example, if the BNRF1 protein is no longer detected at day five following infection of primary B cells and centrosomes are determined to be amplified at day ten following infection, any direct role for the protein in that amplification would be unlikely.

The authors need also to confirm or refute their second conclusion directly. For example, if they express only BNRF1 in RPE1 cells constitutively are these cells rendered more tumorigenic than are the parental cells infected with an empty vector? Without such a direct, confirmatory experiment, their second conclusion is indirect and therefore weak.

Reviewer #3 (Remarks to the Author):

The association between viral infections and cancer development is well established. In particular, EBV infections have been associated with certain types of lymphomas, such as Burkitt's and Hodgkin's lymphomas where high fractions of tumours are EBV positive. Interestingly, although EBV can transform B cells, it is unclear how infected cells become malignant.

In this manuscript, Shumilov et al. investigate possible causes of EBV mediated transformation in B cells. They found that EBV-infected lymphocytes display increased centrosome number, and consequently carry high levels of CIN. In addition, the presence of CIN seems to be associated with tumour frequency in immunocompromised mice. The authors also found that the viral protein BNRF1 is linked to centrosome amplification induced by EBV.

In my opinion, this manuscript makes important contributions to our understanding of EBV-mediated transformation and also highlights the importance of BNRF1 KO virus to produce vaccines to avoid CIN. However, there are several issues that need to be addressed by the authors. The major weakness of the manuscript is the lack of mechanistic insight about the role of BNRF1 in centriole overduplication, which compromises in part the novelty of the results presented. In addition, the in vivo experiments require further clarification as to whether there is a significant difference between cells infected with M81 or M81/ Δ ZR. Below are listed the comments that need to be addressed by the authors:

Major comments:

#1. The major issue of this manuscript is the lack of mechanistic insight about how expression of BNRF1 causes centriole overduplication. There are known players involved in centriole overduplication, such as Plk4, Sas-6, STIL. The authors should assess how BNRF1 protein induces centriole amplification based on what we already know about this process.

#2. In vivo experiments are not clear. The authors claim that there is a difference in tumour development between cells infected with M81 and M81/ Δ ZR. However this is only observed when 4×10^4 cells are injected but not when 4×10^5 cells are injected? Why would this be? This suggests that the effects of CIN in M81 cells are only part of the story?

#3. Figure 2. Panel e: it will be important to know the basal levels of aneuploidy in non-infected cells.

#4. Figure 4. The authors want to claim that centrosome amplification induced by BNRF1 is a major cause of CIN. However, levels of aneuploidy were never measured in the cells infected with BNRF1 KO virus. This is essential to assess whether there are other causes of CIN and aneuploidy in infected cells.

#5. Extended Figure 2. The % of centrosome amplification varies from interphase (around 4%) to mitosis (around 13%). Why is that? Do the authors know whether cells are arresting in interphase? In addition, the % of aneuploidy mitosis is quite high (around 30%) when compared with the % of centrosome amplification in mitosis. The authors should comment on this and show the % of aneuploidy mitosis in cells with and without extra centrosomes. Furthermore, control cell line prior to infection is missing in all graphs.

#6. Extended Figure 4. Panels a-c: CEP170 staining to show the presence of one or more mother centrioles is not clear and looks messy. The authors should also use a centriole marker, such as centrin, instead of gamma-tub. Panel d: quantification of centrosome overduplication should be also shown for the M81/ Δ ZR. Panel e: levels of endogenous Plk4 seem quite high in comparison to what people have reported previously. It would be good to have a control cell line prior to infection as control and evidence that what is being assessed corresponds to Plk4 protein.

Minor comments:

#7. The figures would be properly referenced in the text. It is very difficult to know what the authors mean when each figure has multiple panels and are only referenced in the text as Figure 1, for example, instead of Figure 1a, etc.

#8. Page 4 line 77, extended figure 2 should be extended figure 3?

#9. Page 4 line 84, extended figure 2 should be extended figure 3?

#. Page 4 line 92, extended figure 1 should be extended figure 3? This is unclear because the authors reference this to CIN and aneuploidy levels which in extended figure 1 they are quantifying centriole amplification and mitotic defects, which is not the same.

#10. Page 5 lines 101-103. The authors make a distinction between centriole overduplication and centriole amplification. This is not accurate since centriole overduplication is a cause of centriole amplification. The distinction the authors want to make is between centriole overduplication and for example cytokinesis failure or cell fusion, which would lead to extra centrioles that carry more than one mother.

We are grateful to the referees for their very prompt and detailed review of our paper. We have accommodated their remarks in the revised version of our paper, and performed the suggested experiments wherever feasible.

Reviewer 1

The manuscript by Shumilov and colleagues nicely demonstrates that EBV infection can cause chromosomal instability and associated centrosomal abnormalities soon after infection of B cells. Interestingly, this is caused by a viral tegument protein (BNRF1) that was not considered to be one of the main EBV oncoproteins. These findings have important implications for the way we think about viral oncogenesis, and could explain some EBV-negative cancers where abortive infection can contribute to cellular transformation (and potentially hit-and-run long-lasting effects).

Answer: we thank the referee for these positive comments. We agree about the important implications of the work.

Specific comments

1. Extended data Fig 3: The M81 infected cells showed a recurrent t(6;9) and the ones infected with B95-8 showed a recurrent t(9;15). To me, this suggests early expansion of a single clone in these cultures, and perhaps these translocations should not be considered independent events but rather selection of the clones with a translocation.

Independent infections of different donors followed by karyotyping should be able to clarify whether in long term cultures translocations are more common for M81 than for M81/ Δ ZR.

At the very least, the text could be more clear about whether these translocations were only seen in cell lines, and the time between infection and karyotyping in panels A, B, C and D.

Answer: The revised paper includes the results of the analysis of four pairs of established cell lines infected with M81 or M81/ Δ ZR and of three B cell sample pairs infected with these viruses, but at 6 days post-infection. We have compared the frequency of translocation, deletion and of clonal events (defined by more than 2 identical abnormal mitoses in the same sample) early and late after EBV infection. This analysis showed that the rate of deletion or translocation is very low in infected cells at an early time point and that none of these are clonal events. At a later time point, the rates of deletion, translocation and of clonal events were higher (Page 6 and 7, Supplementary Fig. 2 and 4, page 15, line 306-312). Therefore, we agree with the referee that selection probably takes place in infected cells.

The same analysis showed that the rate of aneuploidy, deletion, translocation is very similar in cells infected with wild type M81 or M81/ Δ ZR and that none of these cell populations showed clonal events at an early time point. At a later time point, the rate of aneuploidy, deletion, translocation and of clonal events were higher in cells infected with wild type M81 than in those infected with the replication-deficient mutant (Page 6 and 7, Supplementary Fig. 2 and 4).

The legend acknowledges that panels C and D are "cell lines", and in panels A and B not translocations were identified, which I assume are shorter-term cultures.

Answer: there seems to be a misunderstanding here. All EBV-infected cells submitted to a M-FISH karyotype reported in the initial version of the paper were established cell lines that were 4 to 6 weeks old. This information has been included in the revised paper (Supplementary Fig. 4).

2. Line 130 states: "...either with the B95-8 wild type virus or with virus-like particles (VLP) derived from M81 that are devoid of viral DNA and cannot establish a chronic infection." Is this correct? Why was VLP from M81 compared to B95-8? Also in figure 3 there does not appear to be data for B95-8, so this may be a typo.

Answer: We initially tested VLPs from both B95-8 and M81 and obtained the same results that are now given in Supplementary Fig. 6. We then focused on M81 VLPs because they can be produced at much higher levels than those from B95-8. This issue is clarified in the revision, see page 9, line 183-190.

3. BNRF1 has been shown to bind DAXX, which is a histone H3 chaperone, and displace ATRX from this complex, a process reported to modify chromatin to promote latent EBV gene expression. ATRX is thought to be involved in chromosomal segregation during mitosis. ATRX mutations have also been reported in different types of cancers. DAXX has other interesting functions, including some in the DNA damage response. It would be useful to include some description (and references) of the known activities of BNRF1 and a brief discussion of how the authors think that BNRF1 could be inducing chromosomal instability.

Answer: We are aware of the other important properties of BNRF1 and thank the referee for mentioning them. The initial version of the paper was very short and did not allow the inclusion of information that did not directly pertain to the presented data. The revised version allows more space and we have now included this information, together with a more general discussion of BNRF1's contribution to EBV-induced aneuploidy (See page 16).

Reviewer 2

The manuscript by Shumilov et al. describes an impressive body of varied research that leads the authors to two conclusions: The BNRF1 gene product of EBV induces amplification of centrosomes when it enters cells and this function contributes to tumorigenesis. The authors suggest a corollary to their second conclusion: That EBV could potentially contribute to tumorigenesis without being retained in the evolving tumor cells. The authors previously have studied this viral gene and found that it is needed for virions to move, upon cellular entry, their nucleocapsid into the nucleus. How a viral protein with this function leads to the detected amplification of centrosomes remains a mystery.

Answer: we thank the referee for these positive comments.

1) The BNRF1 gene product is not synthesized in newly infected cells but is brought into the infected cell in the tegument of the virion. One uncertainty the authors need to address is the timing with which the amplification proceeds in newly infected cells. They monitor its increase during the first six days following infection and show its decline 30 days after infection.

Answer: we are not sure what this referee means with amplification. We surmise that the referee meant centrosome amplification. We have monitored centrosome amplification in 8 sample pairs infected with M81 WT and M81/ Δ ZR between day 3 and day 30 post-infection, including two new time points at day 6 and at day 15. These data, given in Supplementary Fig. 3 show that the centrosome abnormalities decrease regularly from day 3 to day 30 in cells infected with M81/ Δ ZR. Cells infected with wild type M81 also show a decrease in the frequency of centrosome abnormalities until day 15, but these re-increase sharply at day 30 when lytic replication begins (see Supplementary Fig. 3, paper page 6, line 107 to 112).

Given that the BNRF1 protein will only be lost as the infection proceeds, it is imperative for the authors to show that the detected amplification of centrosomes reflects the levels of expression of the BNRF1 protein in the infected cells temporally or their first conclusion is untenable. For example, if the BNRF1 protein is no longer detected at day five following infection of primary B cells and centrosomes are determined to be amplified at day ten following infection, any direct role for the protein in that amplification would be unlikely.

Answer: We showed in the initial submission that centrosomes are already amplified at day 3 post-infection (Fig. 1a, Fig. 1b, Fig. 1c, Fig. 2a, Fig. 2b). We have now monitored BNRF1 protein level in primary B cells infected with wild type M81. We found that the protein is still detectable 120 hours after infection (Supplementary Fig. 10, page 10 last paragraph). Thus, BNRF1 is still present when centrosome over-replication has already begun.

2) The manuscript by Shumilov et al. describes an impressive body of varied research that leads the authors to two conclusions: The BNRF1 gene product of EBV induces amplification of centrosomes when it enters cells and this

function contributes to tumorigenesis. authors need also to confirm or refute their second conclusion directly. For example, if they express only BNRF1 in RPE1 cells constitutively are these cells rendered more tumorigenic than are the parental cells infected with an empty vector? Without such a direct, confirmatory experiment, their second conclusion is indirect and therefore weak.

Answer: Our observations, based on transfection of BNRF1 and infections with a BNRF1-null virus mutant led us to conclude that BNRF1 induces centrosome amplification and aneuploidy in infected cells, not that it directly contributes to tumorigenesis. We showed that EBV particles induce centrosome amplification and CIN in proliferating cells and that infection of B cells with a replication-competent EBV leads to an increased tumorigenicity in an animal model. However, we did not conclude from the ability of BNRF1 to induce centrosome amplification that this protein on its own, independently of this particular cellular context, increases tumorigenicity and this would be probably impossible to demonstrate for the following reasons:

The link between centrosome amplification, CIN and tumor development is far from direct. The literature agrees that CIN is a frequent feature of cancer that probably increases the tumorigenicity, metastatic potential and resistance to cancer treatment (see for example Holland and Cleveland, *EMBO reports* 2012¹; Godinho and Pellmann, *Philosophical transactions of the royal society B* 2014²; Giam and Rancati, *Cell Division* 2015³. Moreover, individuals with genetic syndromes that lead to aneuploidy have an increased risk factor for cancer (ibid). Thus, we concluded that the centrosome amplification induced by the EBV particles is a risk factor for cancer development. However, it is equally clear that the cellular context in which CIN takes place is essential to determine its impact on cell growth. Indeed, CIN can actually lead to cell death and reduce tumorigenicity, in particular in cells that have retained the ability to induce p53-mediated apoptosis (ibid). Therefore, centrosome amplification and CIN are mainly observed in high-grade tumors that have lost all checkpoints and abilities to induce apoptotic mechanisms. In these particular cells, the CIN conferred the plasticity required to adapt to abnormal growth conditions such as those imposed by hypoxia or in the presence of anti-cancer treatment. We have indicated in the discussion of the paper that the latent viral proteins expressed by infected cells most certainly play a crucial role in modulating the response of the cells to the CIN induced by EBV particles (page 15, line 315-323). Consequently, it is clear that the impact of BNRF1 or of the EBV particles on the cellular phenotype will differ according to the cellular context and we have already identified EBV-infected cells as a cellular system in which the contribution of the EBV particles can be properly studied.

This referee requested to transfect RPE-1 cells with a BNRF1 expression vector and to assess whether this increases the tumorigenicity of RPE-1 cells. However, the literature indicates that RPE-1 cells are immortalized with telomerase but are non-malignant cells (Potapova et al. *Cancer Metastasis Rev*, 2013⁴). Therefore, given the aforementioned complex relationship between CIN and cancer development, it is unlikely that BNRF1 on its own could transform RPE-1 cells. If it were the case, it would also contradict the widely accepted multistep model of cancer development.

We have nevertheless performed the experiment suggested by the referee and assessed growth in soft agar of RPE-1 stably infected cells with a lentivirus that expresses BNRF1, relative to cells infected with a control lentivirus. Neither type of cells could grow under these conditions.

We have included some of the aforementioned aspects in the discussion but not the results of the BNRF1 expression in RPE-1 cells, as, for the reasons indicated above, we do not think that the RPE-1 cells represent a suitable model to address the question. Moreover, we have already addressed it in the context of EBV infection.

Reviewer 3

The association between viral infections and cancer development is well established. In particular, EBV infections have been associated with certain types of lymphomas, such as Burkitt's and Hodgkin's lymphomas where high fractions of tumours are EBV positive. Interestingly, although EBV can transform B cells, it is unclear how infected cells become malignant.

In this manuscript, Shumilov et al. investigate possible causes of EBV mediated transformation in B cells. They found that EBV-infected lymphocytes display increased centrosome number, and consequently carry high levels of CIN. In addition, the presence of CIN seems to be associated with tumour frequency in immunocompromised mice. The authors also found that the viral protein BNRF1 is linked to centrosome amplification induced by EBV.

In my opinion, this manuscript makes important contributions to our understanding of EBV-mediated transformation and also highlights the importance of BNRF1 KO virus to produce vaccines to avoid CIN. However, there are several issues that need to be addressed by the authors. The major weakness of the manuscript is the lack of mechanistic insight about the role of BNRF1 in centriole overduplication, which compromises in part the novelty of the results presented. In addition, the *in vivo* experiments require further clarification as to whether there is a significant difference between cells infected with M81 or M81/ Δ ZR. Below are listed the comments that need to be addressed by the authors:

Answer: we thank the referee for his positive comments. We also agree that the manuscript makes important contributions.

We agree that we have not fully elucidated the biochemical mechanisms through which BNRF1 induces centrosome overduplication. However, using multiple genetic systems, we have clarified the contribution of lytic replication to tumorigenesis, one important long-standing question in the pathogenesis of EBV-associated tumors. Furthermore, we have shown that the virions themselves cause this centrosome amplification, a completely new mechanism with important consequences for the understanding of how viruses can increase the risk of cancer development. Finally, we have identified the main viral protein that mediates these effects and shown that it is recruited to the centrosomal compartment. Therefore, we have made substantial advances in the dissection of the mechanisms that lead to CIN after EBV infection.

The *in vivo* experiments are clearly significant under the usual conditions (4×10^4 infected cells per mouse). However, under these conditions, tumors do not appear in all animals treated with M81/ Δ ZR and we could not study their mitotic process and karyotype. Therefore, we injected 10 times more cells than we normally do to increase the frequency of tumors. Even under these unusual conditions, there was a significantly lower tumor cell burden in animals infected with the M81/ Δ ZR mutant, relative to wild type M81.

1) The major issue of this manuscript is the lack of mechanistic insight about how expression of BNRF1 causes centriole overduplication. There are known players involved in centriole overduplication, such as Plk4, Sas-6, STIL. The authors should assess how BNRF1 protein induces centriole amplification based on what we already know about this process.

Answer We agree that we have not fully elucidated the biochemical mechanisms through which BNRF1 induces centrosome overduplication. However, using multiple genetic systems, we have clarified the contribution of lytic replication to tumorigenesis, one important long-standing question in the pathogenesis of EBV-associated tumors. Furthermore, we have shown that the virions themselves cause this centrosome amplification, a completely new mechanism with important consequences for the understanding of how viruses can increase the risk of cancer development. Finally, we have identified the main viral protein that mediates these effects and shown that it is recruited to the centrosomal compartment. Therefore, we have made substantial advances in the dissection of the mechanisms that lead to CIN after EBV infection.

We have performed the experiments requested by the referee and determined the expression levels of Plk4, SAS-6, STIL in 3 pairs of M81 and M81/ Δ ZR LCLs, in combination with additional controls (primary B cells, mitogen-stimulated B cells, and a lymphoma cell line) (Fig. 6 and Supplementary Fig. 5). Moreover, we now show that BNRF1 expressed in 293 cells is preferentially enriched in the centrosome compartment but that this does not markedly affect the expression pattern of PARP1 and NPM1, two proteins also implicated in the control of centrosome duplication (Fig. 9). There was an increase in the expression of the short form of PARP1 in cells that expressed BNRF1, but the significance of this finding is not clear.

2) *In vivo* experiments are not clear. The authors claim that there is a difference in tumour development between cells infected with M81 and M81/ Δ ZR. However this is only observed when 4×10^4 cells are injected but not when 4×10^5 cells are injected? Why would this be? This suggests that the effects of CIN in M81 cells are only part of the story?

Answer: EBV-infected B cells express multiple viral latent proteins and miRNAs that are known to contribute to the transformation process. EBV-associated tumors also display a high level of CIN. Thus, multiple factors contribute to tumor development induced by the virus. However, very little was known before our work about the mechanisms that lead to CIN and about the role of virus lytic replication in tumor development. We show that EBV lytic replication, through the production of virions, accelerates tumorigenic growth of EBV-infected B lymphocytes *in vivo*. However, it is clear that it is not by far the only contribution of the virus to tumor development. After injection of a high number of infected cells, the tumor growth advantage given by the CIN becomes less visible, although the tumor burden remains significantly higher after infection with the replication-competent viruses than after infection with their replication-deficient counterparts (Fig. 4d). Importantly, the experiments performed with a low number of infected cells are much more likely to reproduce the circumstances under which EBV-induces tumors *in vivo* in humans.

3) Figure 2. Panel e: it will be important to know the basal levels of aneuploidy in non-infected cells.

Answer: The resting B cells that we use for transformation experiments do not grow and therefore do not generate mitotic figures that could be analyzed. Reports from the literature estimate that the normal diploid cells missegregate a chromosome once per hundred of cell divisions *in vitro*⁵. We found that B cells whose growth is induced by CD40 ligand and IL4 display approximately 10% aneuploidy mitoses (see also below).

4) Figure 4. The authors want to claim that centrosome amplification induced by BNRF1 is a major cause of CIN. However, levels of aneuploidy were never measured in the cells infected with BNRF1 KO virus. This is essential to assess whether there are other causes of CIN and aneuploidy in infected cells.

Answer: We agree with this referee and have performed this important experiment. We found that the rate of aneuploidy induced by virus-like particles from B95-8 or M81 drops 2.5 to 3.5 times if BNRF1 is lacking (Fig. 8). The observed aneuploidy rate fell to around 10%, the frequency of aneuploidy observed in non-infected CD40 ligand stimulated B cells. We also indicated in the discussion that in order to perform this experiment, we had to expose B cells to an unusually large number of BNRF1-null viruses to obtain a sufficient number of EBV-infected B cells for the analysis, as BNRF1 is required for an efficient B cell infection (page 14, first paragraph). This might possibly partly explain these results. However, we do not exclude that other EBV proteins within the virion also contribute to the observed CIN (page 14, first paragraph).

5). Extended Figure 2. The % of centrosome amplification varies from interphase (around 4%) to mitosis (around 13%). Why is that? Do the authors know whether cells are arresting in interphase?

Answer: Mitoses with centrosome amplification can be readily identified by the presence of more than 2 centrioles at either mitotic pole. However, an interphase cell with 4 centrioles can either be an abnormal cell with centrosome amplification or a normal cell that has already undergone centrosome replication. Therefore, we applied the criteria that are used in the literature, i.e. the presence of more than 4 centrioles, to identify centrosome amplification in interphase cells. However, these criteria clearly underestimate the percentage of interphase cells with abnormal centrosome amplification. This largely explains the discrepancy between the numbers found in interphase cells and in mitotic cells. Owing to the EBV infection, the B cells are not arrested in interphase.

In addition, the % of aneuploidy mitosis is quite high (around 30%) when compared with the % of centrosome amplification in mitosis. the authors should comment on this and show the % of aneuploidy mitosis in cells with and with extra centrosomes. Furthermore, control cell line prior to infection is missing in all graphs.

Answer: The difference between the percentage of cells with aneuploidy and of those that evince centrosome amplification during mitosis is partly explained by the technical requirements for the identification of aneuploid cells that differ from those necessary for the identification of cells with centrosome amplification. As the referee knows, whilst aneuploidy is generally easy to recognize, centrioles sometimes overlap and cannot always be distinguished. Our very careful and conservative estimation of centrosome

amplification undoubtedly leads to an underestimation of the percentage of mitotic cells with centrosome amplification. Moreover, it is possible that when cells with centrosome amplification divide, one daughter cell could receive both an aneuploid genome and a single centrosome. This cell could in principle go through mitosis and be identified as aneuploid without necessarily carrying an abnormal number of centrosomes.

The observation that the deletion of BNRF1 reduces both the rate of centrosome amplification and the rate of aneuploidy also fits with the hypothesis that both events are largely if not exclusively linked. However, we do not exclude that the EBV particles could also induce aneuploidy through an additional mechanism that is independent of the effect of BNRF1 on the centrosomes. These remarks are included in the discussion page 14.

We are not aware of any experimental method that would allow the simultaneous identification of centrosome amplification and aneuploidy in the same cell. Therefore we cannot directly give the percentage of aneuploid cells without centrosome amplification. However, this can be indirectly calculated from the knowledge of the percentage of cells with aneuploidy and of those with centrosome amplification. As already mentioned, resting B cells do not divide and cannot be used as controls for investigation of mitotic cells.

6) Extended Figure 4. Panels a-c: CEP170 staining to show the presence of one or more mother centrioles is not clear and looks messy. The authors should also use a centriole marker, such as centrin, instead of gamma-tub. Panel d: quantification of centrosome overduplication should be also shown for the M81/ Δ ZR.

Answer: We initially used an antibody against gamma-tubulin because this antibody was used in the initial description of CEP170 as a marker of overduplication (Guarguaglini et al. *Molecular Biology of the cells* 2005⁶). We have repeated our investigations with an antibody specific to centrin as requested. We have also included the results of the CEP170 staining for cells transformed with M81/ Δ ZR (Fig. 6a to d).

7) Panel e: levels of endogenous Plk4 seem quite high in comparison to what people have reported previously. It would be good to have a control cell line prior to infection as control and evidence that what is being assessed corresponds to Plk4 protein.

Answer: The intensity of the signals reflects the exposure time. We have repeated the Western blot against Plk4 to show a shorter exposure and have included non-infected cells as requested (Fig. 6e, page 8, line 164-168). We also compared the expression of centrosomal proteins in EBV-infected B cells and in cell lines such as U2OS, RPE-1 and HeLa and see differences (Supplementary Fig. 5).

8) The figures would be properly referenced in the text. It is very difficult to know what the authors mean when each figure has multiple panels and are only referenced in the text as Figure 1, for example, instead of Figure 1a, ect.

Answer: We have modified the references to the figures as requested wherever relevant.

9) Page 4 line 77, extended figure 2 should be extended figure 3?

Answer: The initial extended Figure 2 (now Fig.2) described the frequency of most abnormalities. Therefore, it was correctly cited in the initial version of our paper.

10) Page 4 line 84, extended figure 2 should be extended figure 3?

Answer: The initial extended Figure 2 (now Fig.2) described the frequency of most abnormalities. Therefore, it was correctly cited in the initial version of our paper.

11). Page 4 line 92, extended figure 1 should be extended figure 3?

Answer: The initial extended Figure 1 (now Supplementary Fig.1) described the frequency of mitotic abnormalities in B95-8-infected B cells. The initial extended Figure 3 (now Supplementary Fig. 4) describes the M-FISH characterization of a cell line infected by B95-8. Therefore, it was correctly cited in the initial version of our paper.

This is unclear because the authors reference this to CIN and aneuploidy levels which in extended figure 1 they are quantifying centriole amplification and mitotic defects, which is not the same.

Answer: In Fig. 1 we summarize the total frequency of the different types of abnormal mitotic figures, not taking into account aneuploidy. In Fig. 2 (previously extended Figure 2) we give the details of these abnormalities, and also the frequency of aneuploidy. We agree that this needs to be stated more clearly in the figure legends and we have modified them accordingly (See figure legend of Fig. 1).

12) Page 5 lines 101-103. The authors make a distinction between centriole overduplication and centriole amplification. This is not accurate since centriole overduplication is a cause of centriole amplification. The distinction the authors want to make is between centriole overduplication and for example cytokinesis failure or cell fusion, which would lead to extra centrioles that carry more than one mother

Answer: We agree with the referee and have modified the text accordingly (page 8, line 149). To simplify the discussion, we called centrosome accumulation an increased centrosome number that contains more than one mother centriole, as previously suggested (Cosenza et al., *Chr. Research* 2016⁷)

References

1. Holland AJ, Cleveland DW. Losing balance: the origin and impact of aneuploidy in cancer. *EMBO reports* **13**, 501-514 (2012).

2. Godinho SA, Pellman D. Causes and consequences of centrosome abnormalities in cancer. *Philosophical transactions of the Royal Society of London Series B, Biological sciences* **369**, (2014).
3. Giam M, Rancati G. Aneuploidy and chromosomal instability in cancer: a jackpot to chaos. *Cell division* **10**, 3 (2015).
4. Potapova TA, Zhu J, Li R. Aneuploidy and chromosomal instability: a vicious cycle driving cellular evolution and cancer genome chaos. *Cancer metastasis reviews* **32**, 377-389 (2013).
5. Gordon DJ, Resio B, Pellman D. Causes and consequences of aneuploidy in cancer. *Nature reviews Genetics* **13**, 189-203 (2012).
6. Guarguaglini G, Duncan PI, Stierhof YD, Holmstrom T, Duensing S, Nigg EA. The forkhead-associated domain protein Cep170 interacts with Polo-like kinase 1 and serves as a marker for mature centrioles. *Molecular biology of the cell* **16**, 1095-1107 (2005).
7. Cosenza MR, Kramer A. Centrosome amplification, chromosomal instability and cancer: mechanistic, clinical and therapeutic issues. *Chromosome research : an international journal on the molecular, supramolecular and evolutionary aspects of chromosome biology* **24**, 105-126 (2016).

REVIEWERS' COMMENTS:

Reviewer #1 (Remarks to the Author):

This paper remains an important advance in our understanding of how EBV may promote oncogenesis through a mechanism different from expression of latent proteins. Concerns were addressed, and the paper is now clear and convincing.

Reviewer #2 (Remarks to the Author):

The findings by Shumilov et al. are insightful and interesting. They demonstrate that infection of B-cells by different strains of EBV induces chromosomal instability. They show also that one gene product of the virus is associated with centrosome amplification. These findings are intriguing for multiple reasons including their raising concerns about the study of EBV-immortalized cell lines for genetic analyses.

One surprising finding that the authors should address is their "infection" of RPE-1 and HeLa cells with derivatives of EBV. EBV in other studies infects epithelial cells quite inefficiently yet the current manuscript appears to infect them as efficiently as B-cells in that the effects they observe in the epithelial cells are similar in magnitude as in B-cells. It would be useful for the authors to provide the efficiencies with which they infect these epithelial cells.

Reviewer #3 (Remarks to the Author):

In this revised version of the manuscript the authors addressed most of my concerns. I think this manuscript is now appropriate for publication in this journal.

REVIEWERS' COMMENTS:

Reviewer #1 (Remarks to the Author):

This paper remains an important advance in our understanding of how EBV may promote oncogenesis through a mechanism different from expression of latent proteins. Concerns were addressed, and the paper is now clear and convincing.

Answer: We thank the referee for these positive comments.

Reviewer #2 (Remarks to the Author):

The findings by Shumilov et al. are insightful and interesting. They demonstrate that infection of B-cells by different strains of EBV induces chromosomal instability. They show also that one gene product of the virus is associated with centrosome amplification. These findings are intriguing for multiple reasons including their raising concerns about the study of EBV-immortalized cell lines for genetic analyses.

Answer: We thank the referee for these positive comments.

One surprising finding that the authors should address is their "infection" of RPE-1 and HeLa cells with derivatives of EBV. EBV in other studies infects epithelial cells quite inefficiently yet the current manuscript appears to infect them as efficiently as B-cells in that the effects they observe in the epithelial cells are similar in magnitude as in B-cells. It would be useful for the authors to provide the efficiencies with which they infect these epithelial cells.

Answer: We agree with the referee that the classical infection of epithelial cells is inefficient and we have now provided its efficiency as defined by the presence of the viral genome and the expression of EBV in the target cells (see suppl Fig. 7 n-q , page 10 line 225 to 229). As indicated in the abstract and in the results section, our results show that EBV can interfere with the cell division machinery of a large range of cell types without having to establish a stable infection. As stated by referee 1, this observation significantly increases the range of mechanisms through which EBV can promote oncogenesis.

Reviewer #3 (Remarks to the Author):

In this revised version of the manuscript the authors addressed most of

my concerns. i think this manuscript is now appropriate for publication in this journal.

Answer: We thank the referee for these positive comments.